# Recurrent Reinforcement Learning with Memoroids

**Steven Morad**[1,2], **Chris Lu**[3], **Ryan Kortvelesy**[2], **Stephan Liwicki**[4], **Jakob Foerster**[3],
**Amanda Prorok**[2]
[1]Faculty of Science and Technology, University of Macau, China
[2]Computer Science and Technology, University of Cambridge, UK
[3]Engineering Science, University of Oxford, UK
[4]Toshiba Europe, UK
smorad@um.edu.mo, christopher.lu@exeter.ox.ac.uk, rk627@cst.cam.ac.uk,
Stephan.Liwicki@toshiba.eu, jakob.foerster@eng.ox.ac.uk, asp45@cam.ac.uk

## Abstract

Memory models such as Recurrent Neural Networks (RNNs) and Transformers address Partially Observable Markov Decision Processes (POMDPs) by mapping trajectories to latent Markov states. Neither model scales particularly well to long sequences, especially compared to an emerging class of memory models called Linear Recurrent Models. We discover that the recurrent update of these models resembles a *monoid*, leading us to reformulate existing models using a novel monoid-based framework that we call *memoroids*. We revisit the traditional approach to batching in recurrent reinforcement learning, highlighting theoretical and empirical deficiencies. We leverage memoroids to propose a batching method that improves sample efficiency, increases the return, and simplifies the implementation of recurrent loss functions in reinforcement learning.

## 1 Introduction

Reinforcement learning (RL) traditionally focuses on solving Markov Decision Processes (MDPs), although for many interesting problems the Markov state is hidden. Instead, we receive noisy or ambiguous *observations*, resulting in Partially Observable MDPs. The standard approach to RL under partial observability involves summarizing a sequence of observations into a latent Markov state using a *memory model* or *sequence model*. Commonly used models include RNNs and Transformers.

Training Transformers or RNNs over long sequences is computationally expensive. Instead, prior work often splits these sequences into shorter fixed-length subsequences called *segments* (Figure 1). Using segments adds implementation complexity, reduces efficiency, and introduces theoretical issues. Despite these drawbacks, most prior work and virtually all existing RL libraries follow this segment-based approach. A new class of sequence models, sometimes called Linear Recurrent Models, offers much greater efficiency over long sequences than Transformers or RNNs. We posit that we can utilize these efficient models to do away with segments and their associated drawbacks.

**Contributions** We aim to remove the need for segments in RL. First, we discover that many efficient memory models share an underlying structure reminiscent of *monoids*, a concept from category theory. We propose to extend the monoid into a *memoroid*, a mathematical framework which can represent a large class of efficient memory models. Armed with the memoroid, we propose a new batching method that eliminates the need for segments.

38th Conference on Neural Information Processing Systems (NeurIPS 2024).

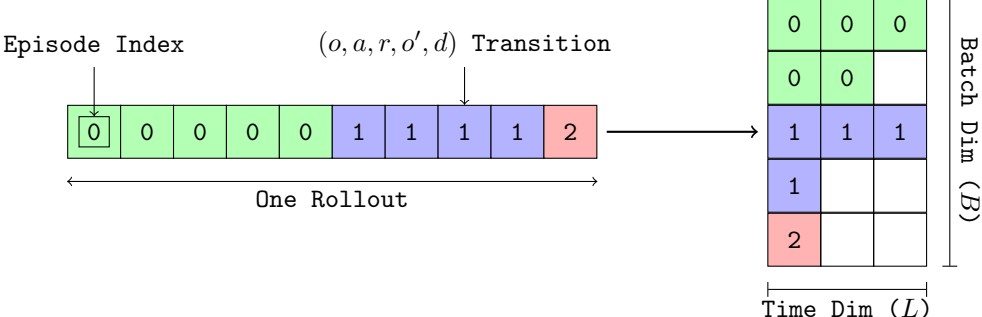

Figure 1: We visualize the Segment-Based Batching approach often used in prior literature. A worker collects a rollout of episodes, denoted by color. Each episode is split and zero-padded to produce a batch of segments, each with a constant, user-specified segment length $L$. Episodes exceeding the specified length are broken into multiple segments, preventing backpropagation through time from reaching earlier segments. Segments contain zero padding, reducing efficiency, biasing normalization methods, and necessitating padding-aware recurrent loss functions.

In particular, we

- Derive memoroids for existing sequence models, as well as the discounted return and advantage
- Introduce a method for inline resets, enabling any memoroid to efficiently process multiple episodes
- Demonstrate that using segments degrades recurrent value functions
- Propose a new memoroid-based batching method that eliminates the need for segments
- Use this batching method to improve sample efficiency and simplify recurrent RL loss functions

## 2 Preliminaries

Consider an MDP $(S, A, R, \mathcal{T}, \gamma)$, where at each timestep $t$, an agent produces a transition $T = (s, a, r, s')$ from interaction with the environment. We let $s, s' \in S$ denote the current and next states and state space, $a \in A$ denote the action and action space, $R : S \times A \times S \mapsto \mathbb{R}$ denote the reward function, and $\mathcal{T} : S \times A \mapsto \Delta S$ denote the state transition matrix ($\Delta$ denotes a distribution). In RL, our goal is to learn a policy parameterized by $\theta$ that maps states to action distributions $\pi_\theta : S \mapsto \Delta A$. The agent samples an action from the policy given the current state $a \sim \pi_\theta(s)$, and stochastically transitions to the next state $s' \sim \mathcal{T}(s, a)$, receiving a reward $r = R(s, a, s')$. The optimization objective is to find the parameters $\theta$ that maximize the expected return, discounted by $\gamma$: $\mathbb{E}_\pi[\sum_{t=0}^\infty \gamma^t R(s_t, a_t, s_{t+1})]$.

### 2.1 Rollouts, Causality, and Episode Boundaries

It is often practical to model terminal states in MDPs, such as a game over screen in a video game. In a terminal state, all actions lead back to the terminal state and the discounted return after entering the terminal state is always zero. We mark whether a state is terminal using the *done flag* $d \in \{0, 1\}$. The done flag is stored in the transition $T = (s, a, r, s', d)$. Transitions are often used in loss functions to train the policy. However, while navigating the MDP we do not have access to the full transition – just the state. We receive the done flag, reward, and next state $(r, d, s')$ at the *next* timestep. This distinction between current and next timestep becomes important when we execute memoroids over multiple episodes.

We find that our paper is more clear if we introduce a *begin flag* $b \in \{0, 1\}$ that is emitted alongside each observation, available during both training and rollouts. The begin flag is 1 at the initial timestep of an episode and 0 otherwise. We differentiate between a transition $T = (s, a, r, s', b, d)$ available only in hindsight, and a partial transition $\overline{T} = (s, b)$ as emitted while navigating the MDP. To reiterate, we can access $\overline{T}$ at any time, but we can only access $T$ during training.

## 2.2 Partial Observability

In partially observable settings, we cannot directly measure the Markov state $s$. Instead, we indirectly measure $s$ via the observation $o \sim \mathcal{O}(s)$, following the observation function $\mathcal{O} : S \to \Delta O$. With the observation replacing the state, interaction with the environment now produces a transition $P = (o, a, r, o', d, b)$ and partial transition $\overline{P} = (o, b)$. For certain tasks, the action from the previous timestep is also necessary, and is implicitly included in the observation.

A sequence of transitions starting where $b = 1$ and continuing until a terminal state is known as an episode $E$. We use a memory model $M$ to summarize the corresponding sequence of partial transitions into a latent Markov state.

$$M : \overline{P}^n \mapsto S^n. \tag{1}$$

If $M$ is recurrent, we may alternatively write $M$ as a single update or batched update respectively

$$M : H \times \overline{P} \mapsto H \times S, \qquad M : H \times \overline{P}^n \mapsto H^n \times S^n, \tag{2}$$

where $H$ is the set of recurrent states.

# 3 Background and Related Work

In deep learning, we often wish to train memory models over batches of sequences. For a single sequence, we can use Backpropagation Through Time (BPTT) (Werbos, 1990). If the sequences differ in length, it is not immediately clear how to efficiently combine them into a batch. Williams & Peng (1990) propose Truncated BPTT (T-BPTT), which enables backpropagation over fixed-length sequences that we call *segments*. T-BPTT is the defacto standard for training memory models in both supervised learning and RL (Hausknecht & Stone, 2015; Kapturowski et al., 2019; Hafner et al., 2023; Bauer et al., 2023; Liang et al., 2018; Raffin et al., 2021; Huang et al., 2021; Serrano-Muñoz et al., 2023; Lu et al., 2022; Ni et al., 2024).

In *Segment-Based Batching* (SBB), we split and zero pad episodes so that they can be stacked into a tensor with batch and sequence length dimensions $B \times L$. Each row in this tensor is a segment $\sigma$ containing exactly $L$ transitions. Episodes longer than $L$ transitions will be split into multiple *fragments*, such that each is at most $L$ transitions. Fragments shorter than $L$ transitions will be zero padded from the right, such that they become exactly length $L$. We call these padded length $L$ fragments *segments*. We must also store a mask $m$ denoting which elements are zero-padding and which are data. The segments and masks are stacked along the batch dimension, creating $B \times L$ matrices for storage and training (Figure 1). We formally define SBB in Appendix C.

**The Shortcomings of Segments** SBB introduces a number of shortcomings. (1) The zero padding and associated masks must be stored, taking up additional space. (2) The zero padding is fed to the memory model, wasting computation on zeros that are discarded during gradient computation. (3) The zero padding also prevents the use of BatchNorm (Ioffe & Szegedy, 2015) and other normalization methods by shifting the mean and variance of input data. (4) The extra time dimension and padding complicates RL loss functions. (5) Most importantly, when SBB splits episodes into distinct segments, we approximate the true BPTT gradient with the T-BPTT gradient.

Let us demonstrate the effect of SBB on the memory model gradient. Assume we have a loss function $\mathcal{L}$ defined over a model parameterized by $\theta$. We define the true gradient of the loss over an episode of length $n$ as $\nabla \mathcal{L}$. In SBB, we split an episode into length $L$ segments. We approximate the gradient over these segments as $\nabla_\sigma \mathcal{L}$

$$\nabla \mathcal{L} = \frac{\partial \mathcal{L}(\theta, (P_0, P_1, \ldots P_{n-1}))}{\partial \theta}, \quad \nabla_\sigma \mathcal{L} = \sum_{j=0}^{\lceil n/L-1 \rceil} \frac{\partial \mathcal{L}(\theta, (P_{jL}, \ldots P_{\min((j+1)L-1, n-1)}))}{\partial \theta}. \tag{3}$$

Under SBB, we compute the gradient independently for each segment. The gradient across segment boundaries is therefore always zero. With zero gradient, it is unlikely that temporal dependencies greater than the segment length $L$ can be learned. In fact, our experiments show that $\nabla_\sigma$ is often a poor approximation of $\nabla$.

**Alternatives to Segments**   We are not the first to realize the drawbacks of SBB. Hausknecht & Stone (2015) store recurrent states in the replay buffer, while Kapturowski et al. (2019) replay the previous segment to generate a "warm" initial recurrent state for the current segment. These methods improve the return, highlighting issues with zero-initialized states, but do not fix the underlying gradient truncation issue. Real Time Recurrent Learning (RTRL) is an alternative to BPTT, but it has $O(n^4)$ time complexity and is thus much slower (Williams & Zipser, 1989). Irie et al. (2024) propose a faster version of RTRL for RL, but the model must be at most one layer deep. Similar to our work, Lu et al. (2024) avoids truncating backpropagation entirely. They find that this results in greater returns, but do not explore *why* this occurs. Furthermore, their method is restricted to on-policy methods and the S5 memory model. Our method extends Lu et al. (2024) to off-policy algorithms and a large majority of efficient memory models.

## 3.1   On the Efficiency of Sequence Models

SBB evolved alongside RNNs in RL (Hausknecht & Stone, 2015), and Transformers to a lesser extent. Such models are only tractable when the sequence length $L$ is small. RNNs rely on the previous recurrent state to compute the following recurrent state, prohibiting parallelism over the time dimension. Thus, RNNs are unable to exploit the parallelism of modern GPUs over the time dimension. Transformers use pairwise attention on the sequence elements, scaling quadratically in space on the length of the sequence.

A recent class of models espouse time-parallel execution while being either linear or subquadratic in space complexity. These models, such as State Space Models, Linear Transformers, Fast Weight Programmers, RetNet, RWKV, Linear Recurrent Units, Gated Impulse Linear Recurrent Networks, and Fast and Forgetful Memory (Gu et al., 2021; Smith et al., 2022; Schlag et al., 2021; Anonymous, 2023; Peng et al., 2023; Orvieto et al., 2023; Martin & Cundy, 2018; Morad et al., 2023b) are sometimes called *Linear Recurrent Models* because they usually (but not always) employ a Linear Time-Invariant (LTI) recurrent state update, which can be computed in parallel over the time axis (Gu & Dao, 2023).

**Monoids**   Prior work on efficient sequence modeling primarily updates the recurrent state using linear functions (Schlag et al., 2021; Gu et al., 2021; Smith et al., 2022; Orvieto et al., 2023). However, works like Blelloch (1990); Martin & Cundy (2018); Morad et al. (2023b) show that it is possible to create efficient models using nonlinear recurrent updates. The key to efficiency is not that updates are linear, as stated in Gu & Dao (2023), but rather that the recurrent update obeys the associative property. More formally, the recurrent update must be a *monoid* (Bourbaki, 1965). Hinze (2004) shows that *all* monoids have time-parallel implementations.

**Definition 3.1.** A tuple $(H, \bullet, e_I)$ is a monoid if:

$$(a \bullet b) = c \qquad a, b, c \in H \qquad \text{The binary operator } \bullet \text{ is closed on } H \qquad (4)$$

$$(a \bullet b) \bullet c = a \bullet (b \bullet c) \qquad a, b, c \in H \qquad \text{The binary operator } \bullet \text{ is associative} \qquad (5)$$

$$(e_I \bullet a) = (a \bullet e_I) = a \qquad a, e_I \in H \qquad \text{There exists an identity element } e_I \qquad (6)$$

where $\bullet$ for a single input $a$ is defined as $(\bullet\, a) = (e_I \bullet a)$.

Any monoid operator $\bullet$ can be computed in parallel across the time dimension using a parallel scan (Appendix J) (Hinze, 2004; Dhulipala et al., 2021). Given a sequence of length $n$, a work-efficient parallel scan known as the Blelloch Scan executes $O(n)$ calls to $\bullet$ in $O(n)$ space to produce $n$ outputs (Blelloch, 1990). With $p$ parallel processors, the parallel time complexity of the scan is $O(n/p + \log p)$. For large GPUs where $n = p$, the parallel time complexity becomes $O(\log n)$.

## 4   Approach

While powerful, standard monoids are restrictive and cannot represent most Linear Recurrent Models in their entirety. Monoids require that the input, output, and recurrent space be identical. In memory models, we often decouple the input space, from the recurrent state space $H$, from the Markov state space $S$ (Equation 2). Consider, for example, a navigation task where the input is an image, the recurrent state $H$ is a latent map representation, and the Markov state $S$ is a set of $x, y$ coordinates of the agent. In search of a more general memory model framework, we extend the monoid into a memory monoid, or *memoroid* for short.

**Definition 4.1.** $((H, \bullet, e_I), f, g)$ constitute a memoroid if $(H, \bullet, e_I)$ defines a monoid and functions $f, g$ are:

$$f : \overline{P} \mapsto H \qquad\qquad \text{Mapping from a partial transition to the right argument of } \bullet \quad (7)$$

$$g : H \times \overline{P} \mapsto S \qquad \text{Mapping a recurrent state and a partial transition to a Markov state} \quad (8)$$

Recall that a partial transition consists of the observation and begin flag $\overline{P} = (o, b)$. The memoroid defines a recurrent memory model (Equation 2) over a sequence of partial transitions to produce recurrent states $(h_0, h_1, \cdots \in H)$ and then compute Markov states $(s_0, s_1, \cdots \in S)$

$$\begin{bmatrix} h_0 & h_1 & h_2 & \dots \\ s_0 & s_1 & s_2 & \dots \end{bmatrix} = \begin{bmatrix} e_I \bullet f(\overline{P}_0) & e_I \bullet f(\overline{P}_0) \bullet f(\overline{P}_1) & e_I \bullet f(\overline{P}_0) \bullet f(\overline{P}_1) \bullet f(\overline{P}_2) & \dots \\ g(h_0, \overline{P}_0) & g(h_1, \overline{P}_1) & g(h_2, \overline{P}_2) & \dots \end{bmatrix}.$$
$$(9)$$

Given $n$ inputs, functions $f$ and $g$ can each be split into $n$ concurrent threads. Recall that monoids have $O(\log n)$ parallel time and $O(n)$ space complexity. Consequently, *all memoroids have $O(\log n)$ parallel time complexity and $O(n)$ space complexity* on the length of the sequence[1].

**Reformulating Existing Sequence Models**  As an exercise in the flexibility of our memoroid, we rewrite the Linear Transformer, the Simplified State Space Model, the Linear Recurrent Unit, and Fast and Forgetful Memory (Katharopoulos et al., 2020; Lu et al., 2024; Orvieto et al., 2023; Morad et al., 2023b) as memoroids in Appendix G. We note that our monoid reformulation of Morad et al. (2023b) improves upon the original, exhibiting better numerically stability by replacing exponentiated cumulative sums with a Blelloch scan.

**Accelerated Discounted Returns**  Memoroids can model other recurrences as well. For example, we can rewrite the discounted return and Generalized Advantage Estimate (GAE) (Schulman et al., 2016) as a memoroids. Reformulating the discounted return and GAE targets as memoroids enables us to compute them in a GPU-efficient fashion, using a high-level framework like JAX (Bradbury et al., 2018). We find that we can compute such quantities orders of magnitude more quickly than the standard approach. We provide the definitions and proofs of these formulations in Theorem D.1 and Theorem E.1.

**Inline Recurrent State Resets**  So far, we have assumed that we operate memoroids over a single episode using the Blelloch Scan. To scan over a batch of variable-length episodes, we could truncate and zero pad sequences such that each is a fixed length (i.e., SBB). However, this introduces the issues explained in Section 3.

Since memoroids are efficient over long sequences, we could consider concatenating individual episodes into one very long sequence, removing the need for padding and truncation. Unfortunately, as the scan crosses episode boundaries, it feeds information from all prior episodes into future episodes, and information from future episodes into preceding episodes.

To resolve this issue, we propose a *resettable monoid transformation*, which prevents information from leaking across episode boundaries. We can apply this transformation to any monoid (or memoroid), to produce a new monoid that respects episode boundaries.

**Theorem 4.2.** *All monoids $(H, \bullet, e_I)$ can be transformed into a resettable monoid $(G, \circ, g_I)$ defined as*

$$G = \{(A, b) \mid A \in H, b \in \{0, 1\}\} \tag{10}$$

$$g_I = (e_I, 0) \tag{11}$$

$$(A, b) \circ (A', b') = ((A \cdot (1 - b') + e_I \cdot b') \bullet A', b \vee b') \tag{12}$$

*For a single episode, the $A$ term output by the operator $\circ$ is equivalent to the output of $\bullet$. Over multiple contiguous episodes, $\circ$ prevents information flow across episode boundaries.*

*Proof.* See Appendix F. □

---

[1] Assuming (1) The binary operator $\bullet$ and $f, g$ are constant-time and constant-space, which is the case for all Linear Recurrent Models listed thus far. (2) Our processor has $n$ parallel threads of execution.

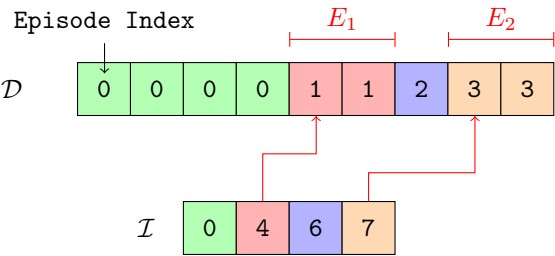

Figure 2: A visualization of sampling in TBB, with a batch size of $B = 4$. Transitions from rollouts are stored in-order in $\mathcal{D}$, with each color denoting a separate episodes. Associated episode begin indices are stored in $\mathcal{I}$. We sample a train batch by randomly selecting from $\mathcal{I}$. For example, we might sample $4$ from $\mathcal{I}$, corresponding to $E_1$ in red. Next, we sample $7$ from $\mathcal{I}$, corresponding to $E_2$ in red. We concatenate $\mathcal{B} = \text{concat}(E_1, E_2)$ and return the result as a train batch.

By transforming the monoid $(H, \bullet, e_I)$ within a memoroid, we no longer require separate time and batch dimensions during training. *Now, memoroids can process a long sequence comprised of many distinct episodes.* Unlike Blelloch (1990); Lu et al. (2024) which provide a reset operator for a specific model, our resettable transformation works for *any* monoid.

### 4.1 Tape-Based Batching

Training over the concatenation of episodes would be intractable using Transformers or RNNs due to poor sequence length scaling, while Linear Recurrent Models leak information between episodes. By combining the efficiency of memoroids with our resettable transform, we resolve these issues, enabling us to fold the batch and time dimensions into a single dimension. We call this approach *Tape-Based Batching* (TBB), which consists of two operations: *insertion* and *sampling*. We provide operations for both on-policy and off-policy RL algorithms. Furthermore, we design TBB in a manner that greatly simplifies the implementation of recurrent loss functions.

**Insertion** In RL, we must store the training data (transitions) that we collect during a rollout. TBB stores them following Algorithm 1. We maintain ordered lists of transitions $\mathcal{D}$, and begin indices $\mathcal{I}$, corresponding to episode boundaries in $\mathcal{D}$. Upon reaching the maximum capacity of $\mathcal{D}$, we discard old transitions by popping the episode begin index from the left of $\mathcal{I}$, and discarding the resulting episode in $\mathcal{D}$. This is guaranteed to discard the oldest episode in $\mathcal{D}$.

This method works both for rollouts that contain complete episodes ($d_{n-1} = 1$), and those that contain incomplete episodes ($d_{n-1} \neq 1$), where a rollout might stop before finishing an episode. When combining incomplete episodes with multiple rollout workers, we can experience race conditions. In this scenario, it is easiest to keep one $\mathcal{D}, \mathcal{I}$ per worker to prevent race conditions.

**Sampling** Once we have constructed $\mathcal{D}, \mathcal{I}$, we are ready to train a policy or compute returns. We sample transitions from $\mathcal{D}, \mathcal{I}$ following Algorithm 2. If we are training on-policy, we can simply train on $\mathcal{D}$. If we are training off-policy, we randomly sample a training batch $\mathcal{B}$ from our dataset by slicing $\mathcal{D}$ using randomly-sampled sequential pairs of episode indices from $\mathcal{I}$. One could extend our sampling approach to implement Prioritized Experience Replay (Schaul et al., 2015) by assigning each episode or index in $\mathcal{I}$ a priority.

**Simplified Loss Functions** With TBB, we utilize unmodified, non-recurrent loss functions to train recurrent policies, reducing the implementation complexity of recurrent RL algorithms. Unlike SBB, there is no need to mask outputs or handle additional time dimensions like in SBB. With TBB, the only difference between a recurrent and nonrecurrent update is precomputing the Markov states $s, s'$ before calling the loss function. We demonstrate this by writing the TBB Q learning update in Algorithm 3, highlighting departures from the standard, non-recurrent Q learning update in red. For posterity, we define the standard SBB Q learning update in Algorithm 4. Note that the SBB update equation has an additional time dimension $k$ and requires a padding mask $m_{i,j}$.

**Algorithm 1** Inserting transitions using TBB

**Input:** List of transitions $\mathcal{D}$, list of indices $\mathcal{I}$, buffer size $D$
**Output:** List of transitions $\mathcal{D}$, list of indices $\mathcal{I}$
$\rho \leftarrow (P_0, P_1, \dots, P_{n-1})$     ▷ Collect rollout from env
**if** on_policy **then**
    $\mathcal{D} \leftarrow \rho$
    $\mathcal{I} \leftarrow \text{where}(b_0, \dots b_{n-1})$     ▷ Indices of new episodes
**else**
    **while** $(\mathcal{D} + \text{card}(\rho)) > D$ **do**
        $\mathcal{I} \leftarrow \mathcal{I}[1:]$     ▷ Buffer full, pop oldest index
        $\mathcal{D} \leftarrow \mathcal{D}[I[0]:]$     ▷ Pop transitions for the oldest episode
    **end while**
    $\mathcal{I} \leftarrow \text{concat}(\mathcal{I}, \text{card}(\mathcal{D}) + \text{where}(b_0, \dots b_{n-1}))$     ▷ Update replay buffer indices
    $\mathcal{D} \leftarrow \text{concat}(\mathcal{D}, \rho)$     ▷ Add new transitions to buffer
**end if**

---

**Algorithm 2** Sampling transitions using TBB

**Input:** List of transitions $\mathcal{D}$, list of indices $\mathcal{I}$, batch size $B$
**Output:** Batch of transitions $\mathcal{B}$
$\mathcal{B} \leftarrow ()$     ▷ Empty list
**while** $\text{len}(\mathcal{B}) < B$ **do**
    $i \sim \mathcal{U}(0, \text{card}(\mathcal{I}) - 1)$     ▷ Randomly sample an index in $\mathcal{I}$
    $\mathcal{B} \leftarrow \text{concat}(\mathcal{B}, D[\mathcal{I}[i] : \mathcal{I}[i+1]])$     ▷ Append episode to batch
**end while**
$\mathcal{B} = \mathcal{B}[:B]$     ▷ Make batch exactly $B$ transitions

---

**Algorithm 3** TBB deep Q update

**Input:** params $\theta$, target params $\phi$, Q function $Q$, sequence model $M$, train batch $\mathcal{B}$, discount $\gamma$, update rate $\beta$
**Output:** params $\theta, \phi$
$(s_1, s_2, \dots s_B) \leftarrow M_\theta(P_1, \dots, P_B)$     ▷ Estimate Markov state
$(s'_1, s'_2, \dots s'_B) \leftarrow M_\phi(P_1, \dots, P_B)$     ▷ Estimate next Markov state
$\hat{y}_j = r_j + \max_{a \in A} \gamma Q_\phi(s'_j, a), \quad \forall \mathcal{B}[j]$     ▷ Compute target
$\theta \leftarrow \min_\theta \|Q_\theta(s_j, a_j) - \hat{y}_j\|, \quad \forall \mathcal{B}[j]$     ▷ Compute loss and update parameters
$\phi \leftarrow \phi\beta + (1-\beta)\theta$     ▷ Update target network params

---

**Algorithm 4** SBB deep Q update

**Input:** params $\theta$, target params $\phi$, Q function $Q$, sequence model $M$, train batch $\mathcal{B}$, discount $\gamma$, update rate $\beta$
**Output:** params $\theta, \phi$

$$\begin{bmatrix} s_{1,1}, & \cdots & s_{1,L} \\ & \vdots & \\ s_{B,1}, & \cdots & s_{B,L} \end{bmatrix} \leftarrow \begin{bmatrix} M_\theta((P_{1,1}) & \cdots & (P_{1,L})) \\ & \vdots & \\ M_\theta((P_{B,1}) & \cdots & (P_{B,L})) \end{bmatrix}$$
▷ Estimate Markov state

$$\begin{bmatrix} s'_{1,1}, & \cdots & s'_{1,L} \\ & \vdots & \\ s'_{B,1}, & \cdots & s'_{B,L} \end{bmatrix} \leftarrow \begin{bmatrix} M_\phi((P_{1,1}) & \cdots & (P_{1,L})) \\ & \vdots & \\ M_\phi((P_{B,1}) & \cdots & (P_{B,L})) \end{bmatrix}$$
▷ Estimate next Markov state

$\hat{y}_{j,k} = (r_{j,k} + \max_{a \in A} \gamma Q_\phi(s'_{j,k}, a)), \quad \forall \mathcal{B}[j,k]$     ▷ Compute target with extra time dimension $k$
$\theta \leftarrow \min_\theta m_{j,k} \cdot \|Q_\theta(s_{j,k}, a_{j,k}) - \hat{y}_{j,k}\|, \quad \forall \mathcal{B}[j,k]$     ▷ Compute loss and update params
$\phi \leftarrow \phi\beta + (1-\beta)\theta$     ▷ Update target network params

## 5   Experiments and Discussion

We begin our experiments by investigating the shortcomings of SBB, specifically the theoretical issues stemming from truncated BPTT. We then compare TBB to SBB across a variety of tasks and models. Finally, we examine the wall-clock efficiency of memoroids.

Our experiments utilize tasks from the POPGym benchmark (Morad et al., 2023a), and all TBB to SBB comparisons use identical hyperparameters and random seeds. We validate our findings across Simplified State Space Models (S5), Linear Recurrent Units (LRU), Fast and Forgetful Memory (FFM), and the Linear Transformer (LinAttn) memoroids. We train our policies using Double Dueling

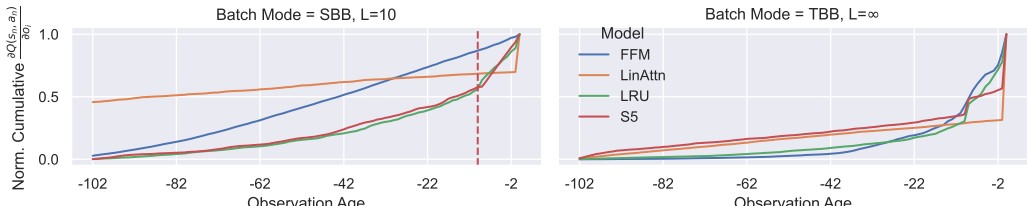

Figure 3: We demonstrate that SBB can hurt Q learning through truncated BPTT. We examine the Repeat Previous task, with RML = 10, comparing SBB (left) to TBB (right). For SBB, we set $L = \text{RML} = 10$ to capture all necessary information. After training, we plot the cumulative partial derivative with respect to the observations on the y-axis. *This partial derivative determines the VML – how much each prior observation contributes to the Q value*. We draw a vertical red line at $L = \text{RML} = 10$. We see that across models, a majority of the Q value is not learnable when using SBB. Even when we set $L = \infty$ using TBB, we see that the VML still spans far beyond the RML. This surprising finding shows that *truncated BPTT degrades recurrent value estimators.*

DQN (Van Hasselt et al., 2016; Wang et al., 2016). See Appendix H for architecture and training details.

**What are the Consequences of Truncating BPTT?**    In Section 3, we discussed how the estimated (truncated) gradient used in SBB differs from the true gradient. We aim to determine whether the estimated gradient used in SBB is a sufficient approximation of the true gradient. We note that if all episodes are a fixed length, and we set $L$ to be this length, both SBB and TBB produce identical results – although this is rare in practice.

We utilize the *Reward Memory Length* (RML) and *Value Memory Length* (VML) metrics from Ni et al. (2024). The RML refers to the maximum temporal dependency $j$ required to predict the expected reward, while the VML determines at which point $k$ prior observations stop affecting the Q value. In other words, the environment defines the RML while the memory model defines the VML. We hope to find that VML = RML; that the Q value only depends on the necessary history.

$$\mathbb{E}\left[R(s, a, s') \mid o_{0:n}\right] = \mathbb{E}\left[R(s, a, s') \mid o_{j:n}\right] \qquad \text{(RML)} \qquad (13)$$

$$Q(M(o_{0:n}), a) = Q(M(o_{k:n}), a) \qquad \text{(VML)} \qquad (14)$$

We examine the VML for the Repeat Previous task with a known RML of ten timesteps. We measure the VML as the impact each observation has on the terminal Q value of an episode (i.e., $Q(s_n, a_n) = R(s_n, a_n, s_{n+1})$). Any observations older than ten timesteps are not necessary to predict the reward, and given a relative-time policy, should have little to no impact on the terminal Q value. Recall that we can write a memory model as $s_n = M(o_0, \ldots o_n)$. We explicitly compute

$$\left|\frac{\partial Q(s_n, a_n)}{\partial o_i}\right| = \left|\frac{\partial Q(s_n, a_n)}{\partial s_n}\frac{\partial s_n}{\partial o_i}\right|, \qquad (15)$$

and plot the results for FFM, S5, LinAttn, and LRU models in Figure 3. We examine the VML and RML of other model and task combinations in Appendix B.

Surprisingly, the VML differs significantly from the RML. The RML is fixed at ten, while the VML appears to span the entire episode. This means that the recurrent models are unable to ignore uninformative prior observations, suggesting that *truncated BPTT degrades recurrent value estimators*. Learned recurrent Q functions do not generalize well over time, although we find that policies trained with TBB tend to generalize better.

In the case of FFM, roughly 90% of the Q value is produced outside the segment boundaries, where truncated BPTT cannot reach. This means that these unnecessary prior observations have a significant impact on our Q values, yet we are unable to forget or ignore such observations. Our findings suggest that SBB could be a major contributor to the increased difficulty and reduced sample efficiency of recurrent RL, as we demonstrate in the next experiment.

**Is TBB More Sample Efficient?**    For our second experiment, we measure the difference in sample efficiency between TBB and SBB. There are two reasons that TBB could improve upon SBB sample

efficiency: (1) As previously discovered, the truncated gradient used by SBB is often a poor estimate of the true gradient (2) SBB decreases the effective batch size through zero padding. We note that the cost of zero padding in SBB is equivalent to the cost of real data – it takes up equivalent space in the replay buffer and takes just as much compute to process as real data. We report some combinations of model and task in Figure 4 and present the full results in Appendix A.

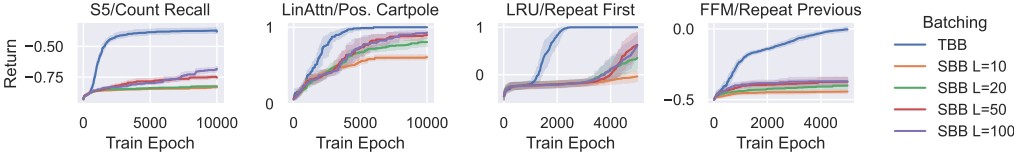

Figure 4: We compare TBB (ours) to SBB across POPGym tasks and memory models, reporting the mean and 95% bootstrapped confidence interval of the evaluation return over ten seeds. We find that TBB significantly improves sample efficiency. See Appendix A for more experiments.

We find that TBB produces a noticeable improvement in sample efficiency over SBB, across nearly all configurations of memory model and environment. Even for large segments lengths $L = 100$, we find a significant gap between SBB and TBB. SBB must make an inherent tradeoff – it can use long segments to improve gradient estimates at the cost of smaller effective batch sizes, or shorter segments to improve the effective batch size at the expense of a worse gradient estimate. TBB does not need to make this tradeoff. In our experiments, SBB with larger $L$ always outperforms shorter $L$, suggesting that the gradient estimation error is a larger contributor to SBB's lackluster performance than reduced effective batch sizes.

**Wall-Clock Efficiency** In Figure 5, we investigate the wall-clock efficiency of memoroids. We find that memoroids compute the discounted return and GAE roughly three orders of magnitude faster than a standard implementation. Next, we compare the wall clock time TBB and SBB take to train a policy from start to finish. For SBB, the parallel time complexity is $O(\log L)$ while TBB has $O(\log B)$ complexity where $B > L$, but in practice there is no perceivable difference in wall-clock time. One possible reason for this discrepancy is that SBB applies expensive split-and-pad operations to the trajectories, while TBB does not. Each fragment contains a varying number of transitions, which corresponds to a varying amount of padding we need to add. Variable-size operations are generally slow and difficult to batch efficiently.

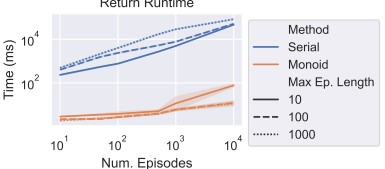

| Method | Train Time (s) | Std. Dev. (s) |
|---|---|---|
| SBB L=10 | 886.39 | 54.47 |
| SBB L=20 | 886.30 | 49.71 |
| SBB L=50 | 887.58 | 50.29 |
| SBB L=100 | 886.25 | 49.87 |
| TBB | 886.87 | 53.21 |

Figure 5: (Left) We compare how long it takes to compute the discounted return using our memoroid, compared to the standard way of iterating through a batch. Computing the discounted return is orders of magnitude faster when using our memoroid implementation. (Right) we compare the total time to train a policy on Repeat First. For both experiments, we evaluate ten random seeds on a RTX 2080Ti GPU.

**Limitations and Future Work** According to our sensitivity analysis, old observations unexpectedly impacted the Q value across models and tasks. Moving forward, we suggest testing newly-designed memory models to see whether VML = RML, to determine whether such models truly generalize over time.

In our experiments, we focused on long-term memory tasks from the POPGym benchmark, each of which tests a specific aspect of long-term memory. We did not experiment on environments like Atari, primarily because it is unclear to what extent Atari tasks require long-term memory.

Although memoroids scale well to long sequences, TBB still pays an increased $\log B$ time cost compared with SBB's $\log L$ cost. There was no perceptible difference in our experiments, but very long sequences such as those used for in-context RL could incur more noticeable training costs. TBB does not strictly require memoroids, but would likely be intractable for RNN or Transformer-based memory models.

## 6    Conclusion

We introduced memoroids as a unifying framework for efficient sequence modeling. We found that memoroids can represent a large number of efficient recurrent models, as well as the discounted return and the advantage. Using our resettable transformation, we extended our approach to encompass batching across variable length sequences. Given the efficiency of memoroids over long sequences, we questioned whether the standard split-and-pad approach to POMDPs was still necessary. We found that said approach causes issues, with shorter segment lengths hampering sample efficiency and ultimately converging to lower returns. We proposed a simple change to batching methodology, that when combined with memoroids, improves sample efficiency at a negligible cost.

## Acknowledgements

We gratefully acknowledge the support of Toshiba Europe Ltd. This work was also supported in part by ARL DCIST CRA W911NF-17-2-0181 and European Research Council (ERC) Project 949940 (gAIa). We thank Matteo Bettini for suggesting the term "memoroid".

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

# A   Return Comparison Between TBB and SBB

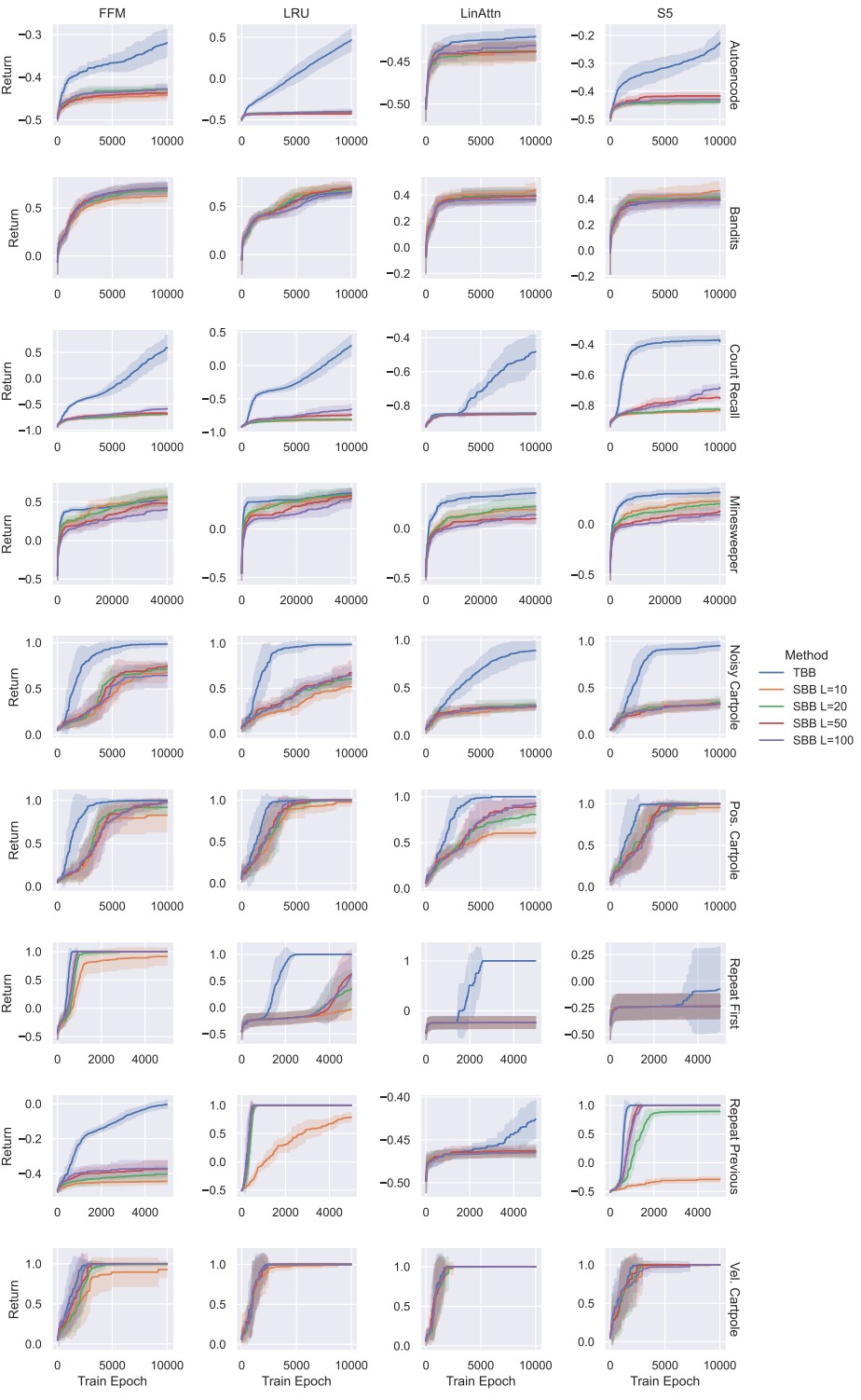

Figure 6: We run four memoroids on nine different POPGym environments over ten seeds and report the mean and 95% bootstrapped confidence interval. The POPGym environments have a minimum episodic return of -1.0 and a maximimum of 1.0. In virtually all experiments, Tape-Based Batching provides improved sample efficiency over all tested segments length using Segment-Based Batching. The Count Recall and Autoencoder environments have temporal dependencies that span the entire sequence, demonstrating the importance of TBB for long range dependencies. On the other hand, Positional Cartpole has a temporal dependency of two timesteps, and so policies trained via SBB can still do reasonably well. Like Count Recall, Repeat First has long term temporal dependencies, however, SBB-trained methods do better than in Count Recall because Repeat First requires storing and recalling only a single observation. All policies perform poorly on bandits because we use a deterministic policy, and this environment benefits from a stochastic policy.

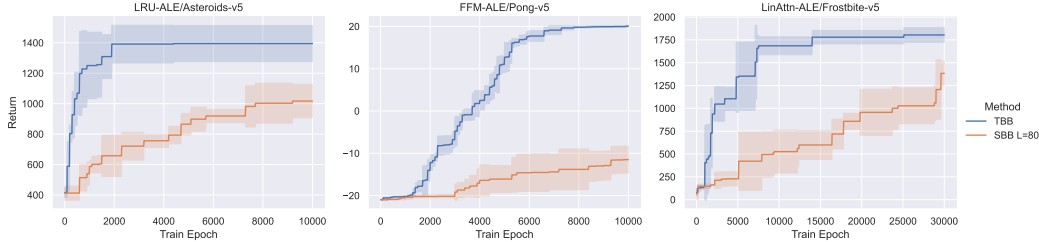

Figure 7: We examine three memoroids on Atari environments from the Arcade Learning Environment (ALE) Bellemare et al. (2013), plotting the mean and 95% confidence interval over three random seeds. In all environments, we see that TBB outperforms SBB.

## B  Observation Sensitivity Analysis

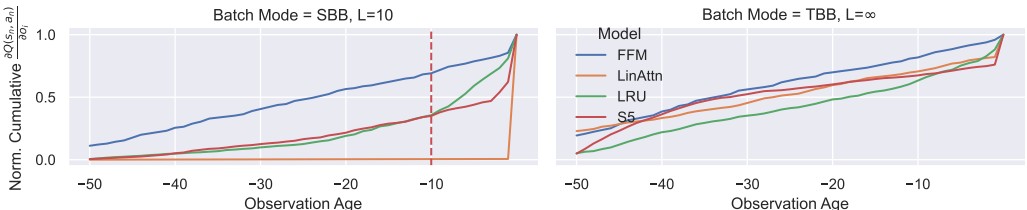

Figure 8: We follow a similar approach to Figure 3 for the Repeat First environment from the POPGym benchmark. In Repeat First, each the agent receives reward for outputting the initial observation at each timestep. Here, we would expect to see virtually all of the probability mass on the initial observation, and zero elsewhere. Again, we see that the gradient is distributed throughout the episode, suggesting that VML will always span the episode. The Linear Transformer (LinAttn) with SBB does very poorly on this task, and so its gradient distribution is not informative.

## C   Segment-Based Batching

After collecting episodes $E$ during a rollout, we split $E$ into fragments $F$ such that each $F$ has a maximum length of $L$. Fragments are zero padded from the right until they are precisely length $L$, turning them into segments $\sigma$ and padding masks $m$. The segments are stacked into a dataset $\mathcal{D}$, enabling easy batching, storage, and training (Figure 1). We define this approach more accurately in the following paragraphs.

We define a segment $\sigma$ as a length $L$ sequence of transitions. During collection, episodes $E$ longer than $L$ transitions are *split* into fragments $F$. Fragments are then *zero-padded* to be length $L$, resulting in fixed-size segments $\sigma$ and associated masks $m$. The resulting segments and masks are stacked into a dataset $\mathcal{D}$, enabling easy batching, storage, and training (Figure 1).

The split function splits a single episode $E$ into one or more fragments $F$, each of size $L$ except for the final fragment.

$$
\begin{array}{cccccc}
F_0 & T_0, & T_1, & \ldots & T_{L-1} \\
F_1 & T_L, & T_{L+1}, & \ldots & T_{2(L-1)} \\
\vdots & = & \vdots \\
F_k & T_{kL}, & \ldots & T_n
\end{array}
\tag{16}
$$

The pad function zero pads a fragment $F$ into a fixed size segment $\sigma$ and associated mask $m$ denoting the padding elements

$$
\sigma, m = \text{pad}(F, L) \tag{17}
$$
$$
= \text{concat}(F, 0^{L-\text{card}(F)}), \, \text{concat}(1^{\text{card}(F)}, 0^{L-\text{card}(F)}) \tag{18}
$$

Using our split and pad operators, we split and pad each incoming episode, producing one or more segments and associated masks for each episode

$$
\begin{bmatrix} \sigma_0, m_0 \\ \vdots \\ \sigma_k, m_k \end{bmatrix} = \text{pad}(F_i, L), \quad \forall F_i \in \text{split}(E, L). \tag{19}
$$

We represent our training dataset $\mathcal{D}$ as the concatenation of segments and masks

$$
\mathcal{D} = \text{concat} \left( \begin{bmatrix} \begin{bmatrix} \sigma_0, m_0 \\ \vdots \\ \sigma_k, m_k \end{bmatrix} \\ \begin{bmatrix} \sigma_{k+1}, m_{k+1} \\ \vdots \\ \sigma_j, m_j \end{bmatrix} \\ \vdots \end{bmatrix} \right) = \begin{bmatrix} \sigma_0, m_0 \\ \vdots \\ \sigma_k, m_k \\ \sigma_{k+1}, m_{k+1} \\ \vdots \\ \sigma_j, m_j \\ \vdots \end{bmatrix} \tag{20}
$$

During training, we randomly sample rows from $\mathcal{D}$ for minibatching (on-policy) or experience replay (off-policy).

# D The Discounted Return as a Memoroid

**Theorem D.1.** *The discounted cumulative return given by*

$$G = \sum_{t=0}^{\infty} \gamma^t r_t \tag{21}$$

*is equivalent to computing the following memoroid over $r_0, r_1, \ldots$*

$$H = \{(a, r) \mid a \in [0, 1], r \in \mathbb{R}\} \tag{22}$$

$$e_I = (1, 0) \tag{23}$$

$$(a, r) \bullet (a', r') = (aa', ar' + r) \tag{24}$$

$$f(o, b) = (\gamma, o) \tag{25}$$

$$g((a, r), (o, b)) = r. \tag{26}$$

*Proof.* We prove the correctness of our discounted return memoroid by showing the expansion is equivalent to the discounted return.

$$(1, 0) \bullet (\gamma, r_0) = (\gamma, r_0 + 0) = (\gamma, r_0) \tag{27}$$

$$(1, 0) \bullet (\gamma, r_0) \bullet (\gamma, r_1) = (1 \cdot \gamma \cdot \gamma, 0 + 1 \cdot r_0 + \gamma r_1) = (\gamma^2, r_0 + \gamma r_1) \tag{28}$$

$$(1, 0) \bullet (\gamma, r_0) \bullet \ldots (\gamma, r_n) = (1 \cdot \gamma \cdot \gamma \cdots \gamma, 1 \cdot r_0 + \gamma r_1 + \ldots \gamma^n r_n) \tag{29}$$

$$= \left(\gamma^n, \sum_{i=0}^{n} \gamma^i r_i\right) \tag{30}$$

If we let $n \to \infty$, we see that the second element in the monoid tuple approaches the discounted return

$$\lim_{n \to \infty} \sum_{i=0}^{n} \gamma^i r_i = \sum_{i=0}^{\infty} \gamma^i r_i \tag{31}$$

$\square$

# E The Generalized Advantage Estimate as a Memoroid

Let us define Generalized Advantage Estimation (GAE) in memoroid form:

**Theorem E.1.** *The GAE target given by*

$$A_t = \sum_{l=0}^{\infty} (\lambda\gamma)^l \delta_{t+l}; \quad \delta_t = r_t + \gamma V(s_{t+1}) - V(s_t) \tag{32}$$

*is equivalent to computing the following memoroid over $\delta_t, \delta_{t+1}, \ldots$*

$$H = \{(a, g) \mid a \in [0, 1], g \in \mathbb{R}\} \tag{33}$$

$$H_I = (1, 0) \tag{34}$$

$$(a, g) \bullet (a', g') = (aa', ag' + g) \tag{35}$$

$$f(o, b) = (\gamma\lambda, o) \tag{36}$$

$$g((a, g), (o, b)) = g. \tag{37}$$

*Proof.* We prove the correctness of our GAE memoroid by showing the expansion is equivalent to the GAE target. This proof is very similar to the proof of the discounted return.

$$(1, 0) \bullet (\gamma\lambda, \delta_t) = (\gamma\lambda, \delta_t + 0) = (\gamma\lambda, \delta_t) \tag{38}$$

$$(1, 0) \bullet (\gamma\lambda, \delta_t) \bullet (\gamma\lambda, \delta_{t+1}) = (1 \cdot \gamma\lambda \cdot \gamma\lambda, 0 + 1 \cdot \delta + \gamma\lambda\delta_{t+1}) = ((\gamma\lambda)^2, \delta + \gamma\lambda\delta_{t+1}) \tag{39}$$

$$(1, 0) \bullet (\gamma\lambda, \delta_t) \bullet \ldots (\gamma\lambda, \delta_{(t+n)}) = (1 \cdot \gamma\lambda \cdot \gamma\lambda \cdots \gamma\lambda, 1 \cdot \delta + \gamma\lambda\delta_{t+1} + \ldots (\gamma\lambda)^n \delta_{t+n}) \tag{40}$$

$$= \left( (\gamma\lambda)^n, \sum_{l=0}^{n} (\gamma\lambda)^l \delta_{t+l} \right) \tag{41}$$

If we let $n \to \infty$, we see that the second element in the monoid tuple approaches the GAE target

$$\lim_{n \to \infty} \left( \sum_{l=0}^{n} (\gamma\lambda)^l \delta_{t+l} \right) = \sum_{l=0}^{\infty} (\lambda\gamma)^l \delta_{t+l} \tag{42}$$

$\square$

# F Resettable Monoid Transformation Proof

*Proof of Theorem 4.2.* First, let us compute all possible pairs of inputs, as we will use them to simplify the rest of the proof.

$$(A, 0) \circ (A', 0) = (A \cdot (1 - 0) + H_I \cdot 0 \bullet A', 0 \vee 0) = (A \bullet A', 0) \tag{43}$$

$$(A, 1) \circ (A', 0) = (A \cdot (1 - 0) + H_I \cdot 0 \bullet A', 1 \vee 0) = (A \bullet A', 1) \tag{44}$$

$$(A, 0) \circ (A', 1) = (A \cdot (1 - 1) + H_I \cdot 1 \bullet A', 0 \vee 1) = (H_I \bullet A', 1) \tag{45}$$

$$\tag{46}$$

Now, we must demonstrate that associativity holds $((A, b) \bullet (A', b')) \bullet (A'', b'') = (A, b') \bullet ((A', b') \bullet (A'', b''))$ for all possibilities of $A, A', A''$ and $b, b', b''$. That is, we must ensure that the episode boundaries are correctly handled for all possibilities – that information does not leak across episode boundaries and that prior information otherwise propagates forward in time.

$$(A \bullet A', 0) \circ (A'', 0) \quad = ((A \bullet A') \cdot (1 - 0) + H_I \cdot 0 \bullet A'', 0 \vee 0) \quad = (A \bullet A' \bullet A'', 0) \tag{47}$$

$$(A \bullet A', 1) \circ (A'', 0) \quad = ((A \bullet A') \cdot (1 - 0) + H_I \cdot 0 \bullet A'', 1 \vee 0) \quad = (A \bullet A' \bullet A'', 1) \tag{48}$$

$$(H_I \bullet A', 1) \circ (A'', 0) \quad = ((H_I \bullet A') \cdot (1 - 0) + H_I \cdot 0 \bullet A'', 1 \vee 0) \quad = (H_I \bullet A' \bullet A'', 1). \tag{49}$$

And for $b'' = 1$, we have

$$(A \bullet A', 0) \circ (A'', 1) \quad = ((A \bullet A') \cdot (1 - 1) + H_I \cdot 1 \bullet A'', 0 \vee 1) \quad = (A \bullet A' \bullet A'', 1) \tag{50}$$

$$(A \bullet A', 1) \circ (A'', 1) \quad = ((A \bullet A') \cdot (1 - 1) + H_I \cdot 1 \bullet A'', 1 \vee 1) \quad = (H_I \bullet A'', 1) \tag{51}$$

$$(H_I \bullet A', 1) \circ (A'', 1) \quad = ((H_I \bullet A') \cdot (1 - 1) + H_I \cdot 1 \bullet A'', 1 \vee 1) \quad = (H_I \bullet A'', 1). \tag{52}$$

We see that resets correctly remove the impact of any terms that occur before $b' = 1$, while correctly propagating state when $b' = 0$. □

# G  Rewriting Sequence Models as memoroids

In this section, we reformulate existing models used in our experiments as memoroids. This reformulation is necessary to use inline resets for these models.

## G.1  Linear Transformer

The Linear Transformer from Katharopoulos et al. (2020) written as

$$X_0 = 0 \in \mathbb{R}^{j \times k} \tag{53}$$

$$x_0 = 0 \in \mathbb{R}^{j} \tag{54}$$

$$X_n = X_{n-1} + \phi(W_k o_n)(W_v o_n)^{\top} \tag{55}$$

$$x_n = x_{n-1} + \phi(W_k o_n) \tag{56}$$

$$s_n = \mathrm{MLP}\left( \frac{X_n\, \phi(W_q o_n)}{x_n^{\top} \phi(W_q o_n)} + o_n \right). \tag{57}$$

can be reformulated as the following memoroid

$$H = \{(X, x) \mid X \in \mathbb{R}^{j \times k}, x \in \mathbb{R}^{j}\} \tag{58}$$

$$e_I = (0, 0) \tag{59}$$

$$(X, x) \bullet (X', x') = (X + X', x + x') \tag{60}$$

$$f(o, b) = (\phi(W_k o)(W_v o)^{\top}, \phi(W_k o)) \tag{61}$$

$$g((X, x), (o, b)) = \mathrm{MLP}\left( \frac{X\, \phi(W_q o)}{x^{\top} \phi(W_q o)} + o \right), \tag{62}$$

where $\phi(x) = 1 + \mathrm{ELU}(x)$.

## G.2  Simplified State Space Models

Prior work (Lu et al., 2024) defines an associate scan operator for the S5 variant of State Space Models. Little work is required to rewrite this in memoroid form:

$$H = \{(X, x) \mid X \in \mathbb{C}^{m \times m}, x \in \mathbb{R}^{m \times 1}\} \tag{63}$$

$$H_I = (I_m, 0) \tag{64}$$

$$(X, x) \bullet (X', x') = (X'X, X'x + x') \tag{65}$$

$$f(o, b) = (W_X, W_x o) \tag{66}$$

$$g((X, x), (o, b)) = (W_1 \,\mathrm{GeLU}(W_c x) + b_1) \odot \mathrm{sigmoid}(W_2 \,\mathrm{GeLU}(W_c x) + b_2) \tag{67}$$

where $W_X, W_x, W_c$ are learnable weights, $b_1, b_2$ are learnable biases, and $I_m$ is the square identity matrix of size $m$.

## G.3  Linear Recurrent Unit

The Linear Recurrent Unit (Orvieto et al., 2023) could be roughly described as a theoretical simplification of S5, bringing it closer to classical RNNs. Writing it out as a memoroid, we see that it nearly identical to S5, however the weight initialization is different

$$H = \{(X, x) \mid X \in \mathbb{C}^{m \times m}, x \in \mathbb{C}^{m \times 1}\} \tag{68}$$

$$H_I = (I_m, 0) \tag{69}$$

$$(X, x) \bullet (X', x') = (X'x, X'x + x') \tag{70}$$

$$f(o, b) = (W_X, W_x o) \tag{71}$$

$$g((X, x), (o, b)) = \mathrm{MLP}(a) \tag{72}$$

## G.4  Fast and Forgetful Memory

Finally, we can rewrite Fast and Forgetful Memory (FFM) as a memoroid, with the parallel scans simplifying its implementation and fixing numerical instabilities caused by large positive exponentials over long sequences, as discussed in Morad et al. (2023b). The original formulation is written as an

aggregator and cell. First, let us write down the $\Gamma$ term used in the aggregator.

$$\Gamma(t) = \exp\left(-t|\alpha|\right)\exp\left(-ti\omega\right)^{\top} \tag{73}$$

$$= \begin{bmatrix} \exp -t(|\alpha_1| + i\omega_1) & \dots & \exp -t(|\alpha_1| + i\omega_c) \\ \vdots & \ddots & \\ \exp -t(|\alpha_m| + i\omega_1) & \dots & \exp -t(|\alpha_m| + i\omega_c) \end{bmatrix} \tag{74}$$

We then write the aggregator as

$$S_{k:n} = \begin{bmatrix} \Gamma(1) \\ \vdots \\ \Gamma(t+1) \end{bmatrix} \odot \begin{bmatrix} S_{k-1} \\ \vdots \\ S_{k-1} \end{bmatrix} + \begin{bmatrix} \Gamma(-t) \\ \vdots \\ \Gamma(0) \end{bmatrix} \odot \begin{bmatrix} \left(\sum_{j=0}^{0} \Gamma(t-j) \odot \left(o_{k+j}1_c^{\top}\right)\right) \\ \vdots \\ \left(\sum_{j=0}^{t} \Gamma(t-j) \odot \left(o_{k+j}1_c^{\top}\right)\right) \end{bmatrix}. \tag{75}$$

where $\odot$ is the Hadamard product (or power), $m$ is the trace size, $c$ is the context size, and $\alpha \in \mathbb{R}^m_+, \omega \in \mathbb{R}^c$ are trainable parameters representing decay and context respectively. Multiplying column a vector by $1_c^{\top}$ "broadcasts" or repeats the column vector $c$ times. The cell is defined as

$$\tilde{x}_{k:n} = \ell_1(o_{k:n}) \odot \sigma(\ell_2(x_{k:n})) \tag{76}$$

$$S_{k:n} = \text{Agg}(\tilde{x}_{k:n}, S_{k-1}) \tag{77}$$

$$z_{k:n} = \ell_3(\text{Flatten}(\Re[S_{k:n}] \,||\, \Im[S_{k:n}])) \tag{78}$$

$$y_{k:n} = \text{LN}(z_{k:n}) \odot \sigma(\ell_4(o_{k:n})) + \ell_5(o_{k:n}) \odot (1 - \sigma(\ell_4(o_{k:n}))). \tag{79}$$

Agg represents the aggregator (Equation 75) and $\ell$ represents linear layers with mappings $\ell_1, \ell_2 : \mathbb{R}^d \to \mathbb{R}^m$, $\ell_3 : \mathbb{R}^{m \times 2c} \to \mathbb{R}^d$, $\ell_4, \ell_5 : \mathbb{R}^d \to \mathbb{R}^d$. $\Re, \Im$ extract the real and imaginary components of a complex number as reals, Flatten reshapes a matrix ($m \times c \to mc$) and $||$ is the concatenation operator. LN is nonparametric layer norm, and $\sigma$ is sigmoid activation. We reformulate and simplify $\Gamma$, the FFM aggregator, and the cell as a single memoroid

$$H = \{(X, t) \mid X \in \mathbb{C}^{m \times c}, t \in \mathbb{Z}\} \tag{80}$$

$$H_I = (0, 0) \tag{81}$$

$$(X, t) \bullet (X', t') = (X \odot \exp\left(t'(-|\alpha| \oplus i\omega)\right) + X', t + t') \tag{82}$$

$$f(o, b) = \left( \begin{bmatrix} (W_1 o + b_1) \odot \sigma(W_2 o + b_2) \\ \vdots \\ (W_1 o + b_1) \odot \sigma(W_2 o + b_2) \end{bmatrix}^{\top}, 1 \right) \tag{83}$$

$$g((X, t), (o, b)) = \text{MLP}(\text{LN}(W_3\left[\Re(X) \,||\, \Im(X)\right] + b_3)) \odot \sigma(W_4 o + b_4) + (1 - \sigma(W_4 o + b_4)) \odot o. \tag{84}$$

where $W, b$ are learnable weights and biases, $\Re, \Im$ extract the real and imaginary part of a complex number, $\odot$ is the elementwise product, $\oplus$ is an outer sum, and $\alpha \in \mathcal{R}^n, \omega \in \mathcal{R}^m$ are learnable parameters. Note that the original FFM formulation requires distributing $\Gamma(-t) = \exp t(|\alpha| + i\omega)$ into the sum. Since $\alpha$ is learned, the real component can grow very large and cause numerical instabilities as it overflows even a double precision float. This is discussed in the limitations section of the original paper. Since our formulation utilizes a Blelloch scan, we can do away with the negative exponent, removing the numerical instability. We note that unlike the other memory models we implemented, FFM is a time-varying recurrence because the recurrent updates depends on $t$.

# H  Experiment Setup

The code necessary to reproduce all of our experiments is available at https://github.com/proroklab/memory-monoids. We used the same model hyperparameters across all experiments. Training hyperparameters, such as number of epochs, varied across tasks. To find hyperparameters, we simply ran many experiments using SBB, the approach used in prior literature. Once we arrived at a good set of hyperparameters, we simply reused them for our TBB method.

## H.1  Compute Used

We ran out of GPU credits early in the paper. We estimate roughly 70% of experiments were run on CPU only, across a number of hardware configurations. Thus, it is not straightforward to arrive at a single number. Users should be able to run at least one seed for each experiment we did, on a reasonable laptop, over approximately one week.

## H.2  Model Setup

We construct our model using blocks. A block contains a linear layer with nonparametric layer normalization and leaky ReLU activation. Observations feed into a block, followed by a memory model, followed by two more blocks. The hidden size of all blocks is 256 dimensions. For the S5 and LRU models, stacked two S5 and LRU layers, resulting in a sum of 512 dimensions of recurrent state (256 per layer). The Linear Transformer and Fast and Forgetful Memory models use just a single layer with 256 dimensions of recurrent state. We use the ADAM optimizer without weight decay.

## H.3  Task Setup

For each task, we selected a replay buffer large enough such that no old observations ever needed to be discarded. Epochs Rand, Train describes the number of episodes we collect randomly, and then the number of training epochs. Polyak $\tau$ determines the target network update rate. Batch Size measures the batch size in transitions for each model update. LR is the learning rate with a linear warmup over a specified number of model updates. The ratio describes the number of episodes collected at each epoch, compared to the number of model updates per epoch. $1 : 2$ means we would perform 2 gradient updates for each 1 episode collected. $\nabla$ Clip corresponds to gradient clipping, where the gradient magnitude is rescaled to at most $\nabla$ Clip. $\gamma$ is the decay term used in MDPs. We use a linear learning rate warmup of 200 updates for all tasks.

| Task | Epochs Rand, Train | Polyak $\tau$ | Batch Size | LR | Ratio | $\nabla$ Clip | $\gamma$ |
|---|---|---|---|---|---|---|---|
| RepeatFirst | 5,000, 5,000 | 0.995 | 1,000 | 0.0001 | 1:1 | 0.01 | 0.99 |
| RepeatPrevious | 5,000, 5,000 | 0.995 | 1,000 | 0.0001 | 1:1 | 0.01 | 0.5 |
| CountRecall | 10,000, 10,000 | 0.995 | 1,000 | 0.0001 | 1:1 | 0.01 | 0.99 |
| PosOnlyCartPole | 10,000, 10,000 | 0.995 | 1,000 | 0.0001 | 1:1 | 0.01 | 0.99 |
| VelOnlyCartPole | 10,000, 10,000 | 0.995 | 1,000 | 0.0001 | 1:1 | 0.01 | 0.99 |
| NoisyCartPole | 10,000, 10,000 | 0.995 | 1,000 | 0.0001 | 1:1 | 0.01 | 0.99 |
| AutoEncode | 10,000, 10,000 | 0.995 | 1,000 | 0.0001 | 1:4 | 0.01 | 0.99 |
| MultiarmedBandit | 10,000, 10,000 | 0.995 | 1,000 | 0.0001 | 1:1 | 0.01 | 0.8 |
| MineSweeper | 10,000, 40,000 | 0.9975 | 1,000 | 0.0001 | 1:1 | 0.01 | 0.99 |

## H.4  Wall-Clock Experiment Details

In the Figure 5 plot, we test the wall-clock efficiency of our discounted return monoid against the standard approach of iterating over episodes in a batch. Both the monoid and standard approach are just-in-time compiled on a GPU, however the standard approach requires a for loop when the episode lengths are not fixed. We sample a batch of episodes, where each episode length is sampled from a discrete uniform distribution between one and a maximum episode length. We find that our memoroid computes the discounted return between three orders of magnitude faster.

Next, we compare TBB and SBB scaling. TBB scales worse than SBB ($O(\log B)$ and $O(\log L)$ respectively, where $B$ is the batch size and $L$ is segment length). We question how this overhead translates to wall-clock training time. In the Figure 5 table, we examine the total time spent training, finding that the time difference is negligible. The memory model forward pass is only a fraction of the time spent at each epoch, with environment sampling, replay buffer sampling (and in the case of SBB, splitting, truncating, and padding sequences) all taking a nontrivial amount of time.

## H.5 Atari Experiment Details

We describe the model and training configuration for the Atari experiments below. We use a CNN similar to that of Mnih et al. (2015), with filter sizes 8, 4, 3 and filter channels 32, 64, 64, and layernorm. The CNN is followed by the recurrent model with recurrent states of size 512, and a two-layer MLP of width 512. We collect one episode per training epoch, and perform 5 gradient updates per epoch. We use a batch size of 16,000 transitions for each update, and evaluate our policy every 100 epochs.

# I Non-Recurrent Q Learning

---

**Algorithm 5** Non-recurrent Q learning update

---

**Input:** params $\theta$, target params $\phi$, Q function $Q$, train batch $\mathcal{B}$, discount $\gamma$
$\hat{y}_j = r_j + \max_{a \in A} \gamma Q_\phi(s'_j, a), \quad \forall \mathcal{B}[j]$ $\qquad\qquad\qquad$ ▷ Q Target
$\theta \leftarrow \min_\theta \|Q_\phi(s_j, a_j) - \hat{y}_j\|, \quad \forall \mathcal{B}[j]$ $\qquad\qquad\qquad$ ▷ Q update
$\phi \leftarrow \phi\beta + (1 - \beta)\theta$ $\qquad\qquad\qquad\qquad\qquad\qquad\qquad\qquad$ ▷ Target update

---

# J A Primer on Scans

In this section, we briefly review scans and associative scans. Generally speaking, we express classical RNNs using scans, and linear recurrent models using associative scans which tend to be more efficient.

## J.1 Scans

A *scan* is an operation over a sequence of elements, often used in tensor processing. We write scans as some function $\bullet$ defined over elements $x_1, x_2, \ldots, x_n$

$$h_n = x_1 \bullet x_2, \cdots \bullet x_n \tag{85}$$

In deep learning, we often formulate an RNN using a scan. Consider, for example, the following simple recurrent network

$$h_n = \sigma(W_h h_{n-1} + W_x x_n) \tag{86}$$
$$y_n = W_y h_n \tag{87}$$

where $\sigma$ represents some nonlinearity, and the $W$ terms are learned parameters. We can define $\bullet$ as

$$h_{n-1} \bullet x_n = \sigma(W_h h_{n-1} + W_x x_n). \tag{88}$$

Thus, we can execute a scan $h_0 \bullet x_1 \bullet x_2 \cdots \bullet x_n$ to compute the recurrent state $h_n$ and output $y_n$. Note that in doing so, we must also compute all intermediate recurrent states $h_1, \ldots, h_n$. This is due to the dependence of $h_n$ on $h_{n-1}$. Let us write out the formula for $h_3$ to demonstrate this dependence

$$h_1 = \sigma(W_h h_0 + W_x x_1) \tag{89}$$
$$h_2 = \sigma(W_h \sigma(W_h h_0 + W_x x_1) + W_x x_2) \tag{90}$$
$$h_3 = \sigma(W_h \sigma(W_h \sigma(W_h h_0 + W_x x_1) + W_x x_2) + W_x x_3). \tag{91}$$

Due to their sequential nature, standard scans tend to be slow on a GPU, since all computations must be executed in sequence.

## J.2 Associative Scans

Certain operators $\bullet$ may exhibit the associative property

$$(x_1 \bullet x_2) \bullet x_3 = x_1 \bullet (x_2 \bullet x_3). \tag{92}$$

When this is the case, we may use *associative scans* instead of scans. Associative scans (alternatively called parallel scans) are generally much faster to execute on a GPU than standard scans. While a scan is $O(n)$ parallel time complexity, a work-efficient parallel scan is $O(\log n)$.

The key idea behind parallel scans is that if the operator $\bullet$ is associative, there is no explicit dependency that requires we execute $\bullet$ in series. Rather, we can we can parallelize computation.
Consider the following expression

$$x_1 \bullet x_2 \bullet x_3 \bullet x_4. \tag{93}$$

If $\bullet$ exhibits the associative property, then we can compute the expression as

$$(x_1 \bullet x_2) \bullet (x_3 \bullet x_4). \tag{94}$$

That is, we can compute the first term $z_2 = (x_1 \bullet x_2)$ independently of $z_4 = (x_3 \bullet x_4)$. Then, we can compute the resulting operator $h_4 = z_2 \bullet z_4$.

This is a naiive associative scan – it executes the binary operator $O(n \log_2 n)$ times. The Blelloch Scan (Blelloch, 1990) produces equivalent outputs to the naiive associative scan, but does so in $O(n)$ calls to $\bullet$. The Blelloch Scan algorithm is relative complex to implement or explain, but fortunately it exists in the CUDA and JAX libraries.

