# OpenReview forum: "Recurrent Reinforcement Learning with Memoroids"
_NeurIPS.cc/2024/Conference — NeurIPS 2024 poster_

### Official Review · Reviewer_hZNa · 2024-07-10

**Soundness:** 3
**Presentation:** 2
**Contribution:** 2
**Rating:** 6
**Confidence:** 3

**Summary:**

This paper introduces the concept of *memoroids*, formalizing a method for computing reset-able episodes with recurrent models that use an associative operation for the latent state update. The authors argue that memoroids can simplify the computation of the loss function and further forego the error in gradient computation introduced by truncation. They compare the method to state-of-the-art segment-based batching and find that it leads to improved performance across a range o sequence models, on RL tasks requiring memory.

**Strengths:**

A intuitive and simple, yet powerful generalization of a range of different computations is presented. The limitations of segment-based batching are introduced in a clear fashion. Investigating the effects of the truncation in BPTT is an interesting and worthwhile endeavor!

**Weaknesses:**

The introduction of the method (TBB) itself is not very clear and somehow confusing. Parallel scans, the sole reason for the frameworks efficiency, are only mentioned. A thorough introduction would be appropriate here. The experimental evaluation should be improved. For example, when investigating the effects of truncation in SBB, investigating more than one environment, and also comparing a range of different truncation horizons would be interesting. It seems like the plots demonstrating the sample efficiency of TBB are truncated - not showing the final performance of all methods. Finally, the explanation of the sample efficiency experiment is missing some crucial details: it seems like L and B are chosen in a way to ensure a fair comparison between TBB and SBB, how was it done?

**Questions:**

- The tradeoff between L and B for SBB is nowhere given formally (did i miss it)?

- Shouldn't TBB and SBB with L larger than the max episode length be comparable in performance since the only difference is the zero padding?

- Figure 4: It seems like the plots are truncated - not showing the final performance of all methods? I was hoping to, at least in the Appendix, find the full plots.

**Limitations:**

The authors are upfront about limitations concerning an increased time cost of the approach compared to segment-based batching.

---

> ### Author Rebuttal · Authors · 2024-07-31
>
> Thank you for taking the time to read our paper and provide feedback. We are happy to hear that we present an "intuitive and simple, yet powerful generalization", and that our research is an "interesting and worthwhile endeavor".
>
> We think your reject rating is quite harsh, given your comments. Hopefully we can clarify some points and reach a shared understanding.
>
> First, we would like to bring up two theoretical contributions that were not listed in your review, which we think could be useful even if you are not sold on TBB!
>
> - We derive memoroids for the discounted return and advantage, achieving $O(\log n)$ complexity compared to standard $O(n)$ implementations. As far as we know, we are the first to model these quantites using a parallel scan.
> - We introduce a method for inline resets, enabling any memoroid to efficiently span multiple episodes
>
> Now, let us address your questions and concerns.
>
> ### Weaknesses
> > The introduction of the method (TBB) itself is not very clear and somehow confusing.
>
> Could you elaborate? We are certainly willing to improve the text if you can help us pinpoint which passages are unclear. Would it be helpful to move Algorithms 1, 2, 3 before their descriptions in the text?
>
> > Parallel scans, the sole reason for the frameworks efficiency, are only mentioned. A thorough introduction would be appropriate here.
>
> We provide a summary of parallel scans on lines 138-142, highlighting [1,2] for further reading.
>
> We already devoted almost half the paper (4 / 9 pages) to background and related work, so we cannot fit an introduction to scans in the main paper. We have added a primer on parallel scans to the Appendix.
>
> > The experimental evaluation should be improved. For example, when investigating the effects of truncation in SBB, investigating more than one environment, and also comparing a range of different truncation horizons would be interesting.
>
> Figure 6 in the Appendix compares TBB to four different truncation horizons, across four models, nine environments, and ten random seeds for a grand total of 1800 experiments.
>
> If you are referring to the VML/RML plots, we also investigate another environment in Figure 7 in the Appendix. The nature of these experiments limits them to environments with a fixed episode length and known RML, which are rare. That said, we believe Figure 6 provides sufficient evidence that TBB is beneficial.
>
> As for different truncation horizons of the RML/VML experiments, we show that even with TBB (infinite horizon), we experience similar behavior to SBB. This implies that varying SBB with larger $L$ would not change much.
>
> > Finally, the explanation of the sample efficiency experiment is missing some crucial details: it seems like L and B are chosen in a way to ensure a fair comparison between TBB and SBB, how was it done?
>
> We feed TBB and SBB the same number of transitions for each update. We have updated the text to clarify this.
>
> For SBB with large $L$, a portion of these transitions will be zero padding. This comparison makes sense given our paper's focus on efficiency, because:
>
> > the cost of zero padding in SBB is equivalent to the cost of real data – it takes up equivalent space in the replay buffer and takes just as much compute to process as real data (lines 260-262)
>
> ### Questions
> > The tradeoff between L and B for SBB is nowhere given formally (did i miss it)?
>
> We address this in lines 257-271. We will quote some of this in answering your next question.
>
> > Shouldn't TBB and SBB with L larger than the max episode length be comparable in performance since the only difference is the zero padding?
>
> Not exactly, here is an excerpt from lines 264-271:
>
> > Even for large segments lengths $L = 100$ we find a significant gap between SBB and TBB. SBB must make an inherent tradeoff – it can use long segments to improve gradient estimates at the cost of smaller effective batch sizes, or shorter segments to improve the effective batch size at the expense of a worse gradient estimate. TBB does not need to make this tradeoff.
>
> > It seems like the plots demonstrating the sample efficiency of TBB are truncated - not showing the final performance of all methods.
>
> Sorry, what do you mean by "final performance"? Are you asking for us to train to convergence? If so, this is not standard in RL. Even the papers that introduce the SAC [3] and PPO [4] algorithms do not train to convergence.
>
> The min/max return in POPGym environments is bounded in $[-1, 1]$, so TBB is already solving many of the tasks in Figure 6. Exceptions include `CountRecall` and `Autoencode`. We have rerun `CountRecall` experiments for longer, and plotted the results in the rebuttal PDF. You can see that two TBB models converge to the optimal return. We are also rerunning the `Autoencode` experiments for longer.
>
> We do not truncate any plots -- we plot the full training duration for all experiments. We trained all tasks for either 5k, 10k, or 40k epochs.
>
> ### References
> [1] Hinze. _An algebra of scans._
>
> [2] Blelloch. _Prefix Sums and Their Applications._
>
> [3] Haarnoja et al. _Soft Actor-Critic: Off-Policy Maximum Entropy Deep Reinforcement Learning with a Stochastic Actor._
>
> [4] Schulman et al. _Proximal Policy Optimization Algorithms._

---

> > ### Comment · Reviewer_hZNa · 2024-08-08
> >
> > > Would it be helpful to move Algorithms 1, 2, 3 before their descriptions in the text?
> >
> > Algorithms 3 and 4 are the part that confuses me. Is the $P$ here also a tuple $(o,b)$? If so, how is the Markov-state mapping $M$ in Algorithm 3 different from Algorithm 4?
> >
> > > We have added a primer on parallel scans to the Appendix.
> >
> > This addresses my concern here adequately.
> >
> > > If you are referring to the VML/RML plots, we also investigate another environment in Figure 7 in the Appendix.
> >
> > This is what I was referring to. I think that Figure 7 does not paint such a clear picture.
> > In general, I feel like this paper would benefit from investigating more thoroughly the reason for SBB's inferiority. For example, comparing magnitudes and principal components of gradients for different $L$ or $B$; Or showing that SBB with $L$ equal to the episode length produces the same updates as TBB.
> >
> > > Exceptions include CountRecall and Autoencode.
> >
> > Honestly, I was expecting that all these environments are supposed to be "solvable" in the sense of being published alongside a model that reaches a reward of 1. After going back to the POPGym paper I realized that this was not the case.
> >
> > > We feed TBB and SBB the same number of transitions for each update.
> >
> > This very sentence is missing in the manuscript. Thank you for clarifying!
> >
> > After reading the rebuttal, I have to admit that my initial judgment was indeed a bit harsh due to some misunderstandings on my part. Therefore, I increase the rating to Weak Accept.

---

> ### Author Response · Authors · 2024-08-08
>
> Thank you for the prompt response. Your feedback has already helped us improve the readability of the paper.
>
> > Is the $P$ here also a tuple $(o, b)$?
>
> Yes, our apologies. It should be $\overline{P}$ and explicitly listed under the algorithm `Input:`. We have updated the text.
>
> >  how is the Markov-state mapping $M$ in Algorithm 3 different from Algorithm 4?
>
> We partially address this in lines 215-222, but these lines are on another page. We will see if we can fit the algorithm and paragraph on the same page. The function $M$ itself is identical between Algorithms 3 and 4. The difference is how we structure the inputs to $M$ (either as one long TBB sequence, or a $B \times T$ SBB batch). Note that this is only possible with the reset operator.
>
> > This is what I was referring to. I think that Figure 7 does not paint such a clear picture. In general, I feel like this paper would benefit from investigating more thoroughly the reason for SBB's inferiority. For example, comparing magnitudes and principal components of gradients for different $L$ or $B$; Or showing that SBB with  equal to the episode length produces the same updates as TBB.
>
> We agree. Originally, we were focused on simplifying recurrent loss functions -- we were not expecting TBB to outperform SBB by so much. The sensitivity study was a late addition.
>
> Surprisingly, we show that the unnecessary dependence on prior inputs is still an issue with TBB, though to a much lesser extent. If we can backpropagate over the whole sequence, why are we still sensitive to prior inputs? To what extent does this occur, in say, Atari games? Is this a property of the objective or target network? The RNN weight initialization? The optimizer? It is not clear.
>
> It was already a challenge to fit memoroids, parallel return computations, inline-resets, and TBB into the same paper while maintaining readability. We suspect that figuring out and fixing the root cause could be a paper on its own.
>
> Thank you for updating your score.

---

### Official Review · Reviewer_M71j · 2024-07-11

**Soundness:** 3
**Presentation:** 2
**Contribution:** 3
**Rating:** 6
**Confidence:** 3

**Summary:**

The authors identify a shared structure in the update rules of linear recurrent models, analogous to monoids. Leveraging this insight, they introduce a mathematical framework called memoroids, which unifies the recurrent update rules of these models. They also derive an inline resettable memoroid, which eliminates the need for segment-based batching in training recurrent reinforcement learning (RL) systems, while preventing information leakage across multiple episodes of varying lengths in long training sequences.

The paper demonstrates that segment-based batching fails to accurately compute the true gradient of multi-episode sequences during training, resulting in degraded algorithm performance. Using their resettable memoroid formalism, the authors propose tape-based batching, which simplifies and improves the accuracy of gradient calculation for recurrent loss functions. They recast four linear recurrent models in their framework and show that this approach, combined with tape-based batching, leads to better convergence and improved sample efficiency in POPGym environments that require memory for problem-solving.

**Strengths:**

1. The introduction of the memoroid formalism, the extension to inline resettable memoroids, and the proposal of tape-based batching are all novel contributions.
2. The formalism is clearly explained and the first sections of the paper are clear to follow.
Segment-based batching, the default for training recurrent RL policies, where a sequence of multi-episode experience is zero-padded and split into smaller segments for training, is clearly introduced. The way that tape-based batching simplifies the loss calculations for recurrent systems is also well-defined.
3. The benefits of recasting four existing linear recurrent models within the memoroid formalism and using tape-based batching are empirically shown by the improvement in sample efficiency and converged episode returns in the experimental section.

**Weaknesses:**

1. The sample efficiency and wall clock efficiency experiments are easy to follow and clearly demonstrate the advantages of the formalism. However, the experiment demonstrating that segment-based batching leads to poor recurrent value estimators is unclear. In general, I feel this experiment subtracts from the flow of the paper. The conclusions drawn from this experiment are also not clearly explained.
2. Only benchmarking on one environment is quite limiting, although I understand that POPGym is specifically made to test memory. Is there another environment in which you expect your method to lead to improved performance due to being able to train on very long sequences with more accurate gradients? Perhaps some tasks in the Atari suite? If so, could you please run some experiments to illustrate this?
3. The paper only shows how off-policy algorithms may be improved and focuses only on discrete action space environments. It is mentioned that other work considers on-policy methods. It would be good to include one set of on-policy experiments as well.

**Questions:**

1. Did the authors ever consider the case where the training segments in segment-based batching is a full-episode rollout? What are the effects of doing this on performance?
2. The authors show that the wallclock time it takes to train a tape-based batching system is the same wallclock time it takes to train a segment-based batching system with varying zero-padded segment lengths. Why do you think this is the case?
3. Could the authors please elaborate on the ‘What are the Consequences of Truncating BPTT’ experiment?

**Limitations:**

Yes, the authors mention the limitations of their method.

---

> ### Author Rebuttal · Authors · 2024-08-01
>
> Thank you for spending time to read and critique our paper. We are happy to hear that you consider our contributions novel. Below, we will respond to your questions and concerns.
>
> ### Weaknesses
> > ...the experiment demonstrating that segment-based batching leads to poor recurrent value estimators is unclear. In general, I feel this experiment subtracts from the flow of the paper. The conclusions drawn from this experiment are also not clearly explained.
>
> We have written a more intuitive tutorial under the "Questions" heading at the bottom of our response. Please let us know if this is helpful, and if so, which parts you think would aid in understanding. We will integrate these parts into the text.
>
> > Only benchmarking on one environment is quite limiting...Is there another environment in which you expect your method to lead to improved performance due to being able to train on very long sequences with more accurate gradients? Perhaps some tasks in the Atari suite?
>
> POPGym is not one environment but a collection of environments. Each one has entirely different observation spaces, action spaces, and transition dynamics which makes them quite diverse. We note that we ran over 1800 experiments.
>
> We have added results from Atari Asteroids to the rebuttal PDF. We find that TBB still shows a noticeable benefit over SBB. We plan to add one more Atari environment as well. We do not have enough compute to run the entire Atari suite.
>
> > The paper only shows how off-policy algorithms may be improved and focuses only on discrete action space environments...It would be good to include one set of on-policy experiments as well.
>
> Algorithms 1,2 work on-policy as well, see line 4 of Algorithm 1. Frankly, many other papers just focus on a single algorithm. For example, the inspiration for our work [1] only evaluates PPO. We believe we have already made significant contributions, and while more experiments are always beneficial, focusing on off-policy algorithms is not an inherent weakness.
>
> ### Questions
> > Did the authors ever consider the case where the training segments in segment-based batching is a full-episode rollout? What are the effects of doing this on performance?
>
> Yes, see lines 265-269, quoted below
>
> > Even for large segments lengths L = 100, we find a significant gap between SBB and TBB. SBB must make an inherent tradeoff – it can use long segments to improve gradient estimates at the cost of smaller effective batch sizes, or shorter segments to improve the effective batch size at the expense of a worse gradient estimate. TBB does not need to make this tradeoff.
>
> certain tasks like `CountRecall` have a maximum episode length less than $100$ timesteps, and we evaluated SBB with $L = 100$.
>
> If both
> 1. All episodes are a fixed length
> 2. The SBB truncation length $L$ is set to precisely this length
>
> then SBB and TBB produce identical outputs.
>
> > The authors show that the wallclock time it takes to train a tape-based batching system is the same wallclock time it takes to train a segment-based batching system with varying zero-padded segment lengths. Why do you think this is the case?
>
> SBB applies additional split-and-pad operations to the trajectories, which are quite expensive. Each subsequence contains a varying number of transitions, which corresponds to a varying amount of padding we need to add. Variable-size operations are generally slow and difficult to batch efficiently.
>
> Furthermore, SBB has $\log L$ complexity while TBB has $\log B$. Generally speaking, $B$ is usually only one or two orders of magnitude larger than $L$, which in log space is not very significant.
>
> > Could the authors please elaborate on the ‘What are the Consequences of Truncating BPTT’ experiment?
>
> Yes, it is a bit complex, but hopefully will clarify an important finding.
>
> Consider a simple recurrent network with some weights $W$ that update recurrent state $h$
>
> $$ h_{t} = h_{t-1} + W x_t$$
>
> We compute the error at $h_t$, which backpropagates through $W$ at each timestep
>
> $$ E = \textrm{error}(h_t, W x_1) +  \textrm{error}(h_t, W x_2), \dots $$
>
> Now, what happens when we truncate BPTT, with a truncation length of $L$? Consider the following:
>
> $$ h_{1} = h_{0} + W x_1 $$
>
> $$ \vdots $$
>
> $$ h_{L + 1} = h_{L} + W x_{L+1} $$
>
> Now, let us compute the error again
>
> $$ E = \textrm{error}(h_{L+1}, W x_2) +  \textrm{error}(h_{L+1}, W x_3), \dots $$
>
> By truncating BPTT, we do not consider the $\textrm{error}(h_{L+1}, W x_1)$ term. $W x_1$ is influencing $h_{L+1}$ and we cannot do anything about it!
>
> In Figure 3, we plot how much $x_t, x_{t-1}, \dots, x_1$ affects the Q value $q_t$. In this task, at each timestep $t$, our policy needs to output $x_{t-9}$. In this case, it makes sense to set $L = 10$ right? If $L = 10$, we will backpropagate through all the necessary observations $x_t, x_{t-1}, \dots x_{t-9}$.
>
> Since we cannot change how $x_{t-10}, x_{t-11}, \dots$ impact $q_t$ (and given that $x_{t-10}, x_{t-11}, \dots$ are not necessary to remember $x_{t-9}$) we would hope that the contribution of $x_{t-10}, x_{t-11}, \dots$ to $q_t$ is zero. That no matter what $x_{t-10}, x_{t-11}, \dots$ are, $q_t$ should not change.
>
> **We find that this is not the case at all**. Even if we use TBB which effectively sets $L=\infty$, the $x_{t-10}, x_{t-11}, \dots$ terms all affect $q_t$. This is a huge issue for SBB and truncated BPTT because we cannot change how $x_{t-10}, x_{t-11}, \dots$ contribute to the recurrent state, while simultaneously showing they have a big contribution to $q_t$. Figure 3 shows that even changing $x_{t-80}$ has a noticable impact on $q_t$. At least with TBB, we can in theory learn to not use the $x_{t-10}, x_{t-11}, \dots$ terms. With SBB, we cannot learn to ignore these terms.
>
> We think that this could be a major reason why training recurrent policies in RL is so difficult.
>
> ## References
> [1] https://openreview.net/forum?id=4W9FVg1j6I

---

> > ### Comment · Reviewer_M71j · 2024-08-11
> >
> > I would like to thank the authors for their detailed response and clarifications and especially for the explanation in the Questions section!
> >
> > It would add to your paper if you could include something similar to this explanation in the final manuscript.
> >
> > I also appreciate running more experiments and for the clarification on the `CountRecall` task and the PopGym benchmark suite.

---

> ### Author Response · Authors · 2024-08-11
>
> Thank you for the response.
>
> In the final paper, we will have:
> - Two Atari experiments
> - Rewritten the `What are the Consequences of Truncating BPTT` paragraphs to be more intuitive, roughly following the above explanation
> - Moved some of the more technical bits of this experiment to the Appendix
> - A sentence explaining why SBB is not faster than TBB
> - A sentence explaining that SBB with fixed length episodes, and $L$ set specifically to the episode length, produces identical outputs to TBB
>
> If we have adequately addressed your concerns, please consider updating your score.

---

### Official Review · Reviewer_WNca · 2024-07-13

**Soundness:** 3
**Presentation:** 2
**Contribution:** 3
**Rating:** 5
**Confidence:** 4

**Summary:**

This paper proposes a new interpretation on how to sample from a buffer of data to avoid the well known trade-offs of truncated BPTT. The final proposed method  interprets a recurrent networks as a monoid, and re-uses ideas developed for linear recurrent networks. This interpretation is time-invariant and enables the parallelization across timesteps better taking advantage of how current computer architectures work (i.e. GPUs). The new method for sampling from the experience replay is then compared to a method which segments the episodes into set truncated sequences (i.e. the sequences, termed "segments" here, are decided when data is put into the experience replay buffer). The comparison uses the POPGym benchmark.

**Strengths:**

- The paper addresses an important problem in partially observable domains, i.e. how do we deal with the limitations of truncated BPTT without incurring significant computational costs and variance costs of full BPTT.
- Restricting the recurrent network to be a time-invariant kernel is an interesting constraint not yet explored in the context of reinforcement learning. This is novel, and the motivation for such a study is well received.
- I think the overall approach is well founded, and doesn't have any technical flaws.

**Weaknesses:**

While I think the paper's novel contributions have interest for the RL community looking into partial observability, the language is overly dower towards previous works and the interpretation of prior methods is flawed. I will justify these claims below. I encourage the authors to reconsider their tone for subsequent submissions.


- 1. **Segment Based Batching**
The assertion that reinforcement learning using deep recurrent networks (i.e. DRQN and following results) use what is termed as Segment-Based Batching (SBB) is fundamentally flawed. From (Hausknecht and Stone, 2015) to later works, "segments" are not pre-determined when putting episodes into the buffer. As stated in the **Stable Recurrent Updates** section of (Hausknecht and Stone, 2015) they worked on two approaches. The method they chose to follow randomly samples a starting point in the episode to generate a sequence, initializing the hidden state to zero. Later (Kapturowski et al, 2019) study this as well, albeit in the distributed RL setting. They also randomly sample starting episodes and studied whether to store hidden states from when the sample was gathered or allow for a "burn-in" period. I am unaware of which paper the SBB interpretation comes from as I understand it, and have not seen it in the literature.

Because of this interpretation, and not testing other replay sampling methods (i.e. those presented by (Kapturowski et al, 2019), it is difficult to say for certainty where TBB fits into the literature in terms of performance. I don't even care if TBB beats other approaches, but it should be compared with what the rest of the field actually uses.

- 2. **Tone**

The issue with the tone of the **Background and Related Work** section continues, and wrongfully attributes the core assumption of truncated BPTT (i.e. $\nabla \approx \nabla_\sigma$) to reinforcement learning papers. This algorithm as far as I know was officially first discussed in (Williams and Peng, 1990), and the biased nature of the algorithm (i.e. due to truncating sequences) has been a consistent focus of study since then. While RL papers should be clearer about this assumption they are taking advantage of (as most recurrent architectures have before them), the burden for them to justify the usefullness of this algorithm for training recurrent networks at the time is not really their concern. It has only been recently, with the structured state spaces and linear recurrent networks lines of research, that there have been reasonable alternatives to truncated BPTT.




[(Williams and Peng, 1990)]( https://direct.mit.edu/neco/article-abstract/2/4/490/5561/An-Efficient-Gradient-Based-Algorithm-for-On-Line)


- 3. **Some missing Related Works**

While coming from a different direction, there has been activity in removing the need for truncated BPTT--and sometimes BPTT entirely--in the RL setting. I'll only list a few papers which might be of interest:
- Restricts the state of an RNN to be predictions and mostly removes the need for BPTT in simple domains. This paper also discusses the history of predictive representations as an alternative to traditional recurrent architectures extensively: https://www.jair.org/index.php/jair/article/view/12105. There are several works in that area to choose from.
- From the angle of classical conditioning, this paper looks directly at the effects of truncation on prediction in a sequential decision making process: https://journals.sagepub.com/doi/full/10.1177/10597123221085039

**Questions:**

- Q-1. According to Algorithm 2, if the length of an episode is longer than the chosen batch size, wouldn't TBB also truncate the gradients, or not ever reach the end of the episode? Or is there some way I'm misinterpreting the algorithm?

**Limitations:**

Yes.

---

> ### Author Rebuttal · Authors · 2024-08-06
>
> Thank you for taking the time to read our paper and provide useful feedback. We are glad to hear that our work "addresses an important problem" and that our "overall approach is well founded, and doesn't have any technical flaws."
>
> Let us address your concerns and questions below.
>
> ### Weaknesses
> > Segment Based Batching The assertion that reinforcement learning using deep recurrent networks (i.e. DRQN and following results) use what is termed as Segment-Based Batching (SBB) is fundamentally flawed. From (Hausknecht and Stone, 2015) to later works, "segments" are not pre-determined when putting episodes into the buffer... Later (Kapturowski et al, 2019) study this as well, albeit in the distributed RL setting. They also randomly sample starting episodes and studied whether to store hidden states from when the sample was gathered or allow for a "burn-in" period. I am unaware of which paper the SBB interpretation comes from as I understand it, and have not seen it in the literature.
>
> SBB is equivalent to the approach from Kapturowski et al. with a burn-in length of zero and a zero-initialized recurrent state (Figure 1c in Kapturowski et al.). **Kapturowski et al. does not store full trajectories**. They **do not randomly sample a start point** within the sequence. From Kapturowski et al.:
>
> > Instead of regular (s, a, r, s0) transition tuples, we store fixed-length (m = 80) sequences of (s, a, r) in replay, with adjacent sequences overlapping each other by 40 time steps
>
> In Kapturowski et al., Fig 1c they:
> 1. Split a collected sequence into length 80 segments
> 2. Place these segments in the buffer
> 3. Randomly sample full segments from this buffer
> 5. Initialize the RNN with a recurrent state of zeros
> 6. Burn-in for 0, 20, or 40 timesteps, starting from the beginning of the segment
> 7. Train on the remaining 80, 60, or 40 timesteps
>
> See the R2D2 implementation from Google Research [1], where an "unroll" refers to the fixed 80-timestep sequence.
>
> Indeed, in practice many follow the SBB approach, forming $B \times T$ subsequences so that subsequences can be stacked along the $B$ axis. Examples include
>
> - Seed RL [1]
> - SKRL [2]
> - TorchRL [3]
> - POMDP-Baselines [4]
> - R2L [5]
>
> Few libraries actually implement a nonzero burn-in length. Of the above listed libraries, only TorchRL and SeedRL implement burn-in.
>
> > it is difficult to say for certainty where TBB fits into the literature in terms of performance
>
> TBB is both simpler to implement and computes a strictly more accurate gradient than prior works using burn-in or stored recurrent states. The wall-clock time difference between SBB and TBB is imperceptible. If using a memoroid, we cannot see any reason to choose burn-in or stored-state approaches over TBB.
>
> > The issue with the tone of the Background and Related Work section continues, and wrongfully attributes the core assumption of truncated BPTT (i.e. $\nabla \approx \nabla_\sigma$ ) to reinforcement learning papers. This algorithm as far as I know was officially first discussed in (Williams and Peng, 1990), and the biased nature of the algorithm (i.e. due to truncating sequences) has been a consistent focus of study since then.
>
> The shortcomings of truncation are not unique to RL, as you state. We have added the (Williams and Peng, 1990) citation and a few sentences explaining that this was common in supervised learning as well.
>
> > While RL papers should be clearer about this assumption they are taking advantage of (as most recurrent architectures have before them), the burden for them to justify the usefullness of this algorithm for training recurrent networks at the time is not really their concern. It has only been recently, with the structured state spaces and linear recurrent networks lines of research, that there have been reasonable alternatives to truncated BPTT.
>
> We argue that authors are responsible for how they train their models. With classical RNNs, there is no good alternative to truncated BPTT, hence its widespread use. The authors made the approximations they needed for tractability -- that is perfectly fine. But are you suggesting we should not discuss the shortcomings of prior work? How are we to motivate our contributions without exploring the shortcomings of prior work?
>
> > Some missing Related Works
> > While coming from a different direction, there has been activity in removing the need for truncated BPTT–and sometimes BPTT entirely–in the RL setting. I'll only list a few papers which might be of interest:...
>
> Thank you for the references, we have added both works to the "Alternative to Segments" subsection.
>
> ### Questions
> > Q-1. According to Algorithm 2, if the length of an episode is longer than the chosen batch size, wouldn't TBB also truncate the gradients, or not ever reach the end of the episode? Or is there some way I'm misinterpreting the algorithm?
>
> Yes, we guarantee at most one truncated episode per batch. The key point is that the user need not make the tradeoff between $B$ and $T$ dimensions. For comparison, we are currently using a batch size of 16,000 on Atari which fits in 20GB of GPU memory. We did not need to truncate any episodes during training.
>
> ## References
>
> [1] https://github.com/google-research/seed_rl/blob/master/agents/r2d2/learner.py#L391-L392
>
> [2] https://github.com/Toni-SM/skrl/blob/636936f3ac49c6d2260bd130d72b789ca6dfe42b/skrl/agents/torch/td3/td3_rnn.py#L226
>
> [3] https://github.com/pytorch/rl/blob/da898261ba18cb221a0c3b54a679b692c3610f06/torchrl/objectives/value/utils.py#L214
>
> [4] https://github.com/twni2016/pomdp-baselines/blob/28cb5dda93d8182fbe3e3055b39f839b1274aa94/policies/models/recurrent_actor.py#L112
>
> [5] https://github.com/siekmanj/r2l/blob/247e1cbde3e5fdc2c1c72e31c52b3d08c9d68cf1/algos/off_policy.py#L77

---

> ### Comment · Reviewer_WNca · 2024-08-08
>
> Thank you for the comments! How to use the replay buffer is a deeply understudied part of using recurrent networks in Deep RL, and I want this paper to be successful. While I laid some heavy criticism initially (some of which I think is still valid), I do believe the ideas presented here are good ideas (that is not in dispute). This conversation is around how to make them as impactful as they can be.
>
> The authors are correct. I was mistaken and had forgotten that statement about Kapturowski et al, and missed it when writing this review. While Kapturowski et al do do this, other papers such as Hausknecht and Stone (and several following papers linked below) follow the random update strategy I initially described. While there is room to compare to SBB, I believe there is a need to also compare with the random update strategy (Kapturowski also doesn't really do this comparison in a straightforward manner i.e. afaict there is no ablation showing random starting points vs SBB). This part of the architecture does tend to be under-reported, so maybe it is worth going through and trying to uncover what the choices are in the background sections of this work. I don't think relying on open-source package's implementations as a way to decide what algorithms to compare against is a valid decision.
>
> Some other papers using random updates, but others exist:
> - https://arxiv.org/pdf/1704.07978
> - https://arxiv.org/pdf/2110.05038: You link the codebase of this paper. On inspecting the sequence replay buffer code, I do believe they use the random strategy. You are correct they don't do burn-in though.
> - I was about to link to Impala (https://arxiv.org/pdf/1802.01561), but their explanation of their batching strategy is sufficiently obscured that it could be either. I don't have time to dig in to the code (https://github.com/google-deepmind/scalable_agent/), but might be another example.
>
> I think there was a misunderstanding of my critic of the language surrounding prior work. While I agree we need to be critical of previous approaches, there is a difference between stating weaknesses of the work in a constructive manner and implying previous work was misleading/dishonest. Language like "Prior work <> assumes $\nabla \approx \nabla_\sigma$ but we are not  aware of any theoretical justification..." is not generous when understanding prior work, and to me implies the previous work mislead their readers. While I agree, it would be better to clearly state the underlying assumptions we are making in our algorithms, the language of this paper suggests [Hausknecht and Stone] and following papers were the first to make such an assumption. There are clear reasons why the assumption was made in context of the literature, and is a well known assumption made throughout the use of recurrent models in all of machine learning. A better way to approach this would be to lay out the history of the assumption, and note that several of the RL works use this assumption as well.
>
> I have also updated my scores/review to reflect my misremembering of Kapturowski. I think this paper could be accepted now, but would be more impactful taking into account some of the other weaknesses.

---

> ### Author Response · Authors · 2024-08-08
>
> Thank you for the prompt reply and insightful feedback.
>
> While TBB is certainly useful, it is only one of our contributions. We argue that the unifying memoroid formalism, inline resets, and $O(\log n)$ advantage and return operations are probably more useful to a broad range of readers than TBB. Yes, comparing TBB to the random update would be interesting. But we argue that the current paper is sufficiently useful for the community at large.
>
> Let us discuss the tone next. The papers by (Hausknecht & Stone, 2015; Kapturowski et al., 2019; Igl et al., 2018; Ni et al., 2024) are foundational works and we did not mean to disparage them. We have deleted
>
> > Prior work (Hausknecht & Stone, 2015; Kapturowski et al., 2019; Igl et al., 2018; Ni et al., 2024) $\nabla \approx \nabla_\sigma$, although we are not aware of any theoretical justification for this assumption, as the error between the true and approximated gradient is unbounded. In fact, our experiments show that $\nabla_\sigma$ is often a poor approximation of $\nabla$.
>
> as these sentences are not necessary, given that:
> 1. We now have a few sentences at the start of Section 3 discussing (Williams and Peng, 1990) as a necessity for tractability in deep learning (not just RL)
> 2. The following paragraph already explains the (Hausknecht & Stone, 2015; Kapturowski et al., 2019) approaches more thoroughly and in a better light -- as _solutions_ to the gradient truncation problem.
>
> We hope we that have addressed at least some of your concerns. If we did, we hope that you reconsider your reject rating.

---

### Official Review · Reviewer_KMv5 · 2024-07-13

**Soundness:** 3
**Presentation:** 4
**Contribution:** 3
**Rating:** 7
**Confidence:** 4

**Summary:**

The authors present a novel approach to recurrent reinforcement learning aimed at improving efficiency and performance. They introduce the concept of memory monoids, algebraic structures used to represent and manipulate the memory of RL agents with recurrent components. They also rewrite Simplified State Space Models (S5), Linear Recurrent Units (LRU), Fast and Forgetful Memory (FFM), and the Linear Transformer (LinAttn) as memoroids, demonstrating that memoroids can represent a large class of sequence models. While the method is slightly more computationally expensive compared to existing methods, it promises higher sample efficiency and improved performance.

**Strengths:**

- The paper is very well-written. It was a pleasure reading this work!
- Clear experiments that demonstrate the superiority of tape-based batching to segment-based batching
- Fig.3 discusses the consequences of truncating BPTT, which is quite interesting
- Memorids can offer GPU efficient computation for existing recurrent variables like $\lambda-$return. Its simple enough to reformulate them

**Weaknesses:**

- The experiments are carefully chosen to showcase certain features of memoroids. That said, they are fairly simple tasks which do not provide any insight about how well memoroids would perform on more challenging tasks
- The authors might want to consider more complex envs like Memory MAze (https://github.com/jurgisp/memory-maze) to really showcase the strength of their method

**Questions:**

- The authors mention that "We did not experiment on environments like Atari, primarily because it is unclear to what extent Atari tasks require long-term memory." I agree with this argument. I'd still be curious about games requiring counting or tracking objects across frames. For example, *Frostbite: The agent needs to keep track of ice floes and build igloos over time or Asteroids: Remembering the positions and trajectories of multiple asteroids can be crucial.*  Do the authors have any guesses for how their approach would be beneficial here?

- "TBB does not strictly require memoroids, but would likely be intractable for RNN or Transformer-based memory models." This is mentioned in the limitations by the authors. I'm curious about how they'd handle tasks that would require RNNs or Transformers.

**Limitations:**

I agree with the limitations listed by the authors.

---

> ### Author Rebuttal · Authors · 2024-08-01
>
> Thank you for reviewing our paper, we are happy to hear that it was a pleasure to read!
>
> > The experiments are carefully chosen to showcase certain features of memoroids. That said, they are fairly simple tasks which do not provide any insight about how well memoroids would perform on more challenging tasks
> The authors might want to consider more complex envs like Memory MAze (https://github.com/jurgisp/memory-maze) to really showcase the strength of their method
>
> > The authors mention that "We did not experiment on environments like Atari, primarily because it is unclear to what extent Atari tasks require long-term memory." I agree with this argument. I'd still be curious about games requiring counting or tracking objects across frames. For example, Frostbite: The agent needs to keep track of ice floes and build igloos over time or Asteroids: Remembering the positions and trajectories of multiple asteroids can be crucial. Do the authors have any guesses for how their approach would be beneficial here?
>
> In our rebuttal PDF, we have added results for Atari Asteroids. We were surprised to find that TBB provides a significant improvement over SBB, even on tasks with weak partial observability. We plan to test Frostbite next.
>
> If accepted, we will add a Memory Maze task as well.
>
> > "TBB does not strictly require memoroids, but would likely be intractable for RNN or Transformer-based memory models." This is mentioned in the limitations by the authors. I'm curious about how they'd handle tasks that would require RNNs or Transformers.
>
> For RNN or Transformers, one could just use more GPUs. Transformers in particular are strictly limited to shorter sequences due to their $O(n^2)$ space complexity. One solution could be to use strong regularization, so that we can significantly reduce the batch size/sequence length.

---

> > ### Comment · Reviewer_KMv5 · 2024-08-08
> > **Thank you for the response**
> >
> > I would like to thank the authors for answering my questions. I’m particularly pleased with their experiments on Atari asteroid, which reassures my rating of Accept. I understand that experiments on Frostbite and memory maze would require more forethought and run times. I’m looking forward to seeing them in the final manuscript.

---

### Author Rebuttal · Authors · 2024-08-06

We thank the AC and all the reviewers for taking time to read our paper and provide useful feedback. In general, the reviewers consider our contributions beneficial:

`KMv5` writes
> The paper is very well-written. It was a pleasure reading this work!

> Clear experiments that demonstrate the superiority of tape-based batching

`WNca` writes
> The paper addresses an important problem

> This is novel, and the motivation for such a study is well received

> the overall approach is well founded, and doesn't have any technical flaws

`M71j` writes
> The introduction of the memoroid formalism, the extension to inline resettable memoroids, and the proposal of tape-based batching are all novel contributions

> The formalism is clearly explained and the first sections of the paper are clear to follow

> The benefits...are empirically shown by the improvement in sample efficiency and converged episode returns

`hZNa` writes
> A intuitive and simple, yet powerful generalization of a range of different computations is presented

> Investigating the effects of the truncation in BPTT is an interesting and worthwhile endeavor!

We have attempted to address each reviewer's concerns individually. In particular, we have added an Atari pixel task and POPGym task to the rebuttal PDF for reviewers `KMv5`, `M71j`, and `hZNa`. This brings the total number of experiments to over 2,000.

We have also agreed to add an appendix section introducing parallel scans for reviewer `hZNa`, and additional citations for reviewer `WNca`.

---

### Decision · Program_Chairs · 2024-09-25

**Decision:**

Accept (poster)

**Comment:**

Dealing with partial observability is one of the most importance challenges in reinforcement learning. This paper proposes a memory-based architecture that makes memory computation efficient while simplifying the loss function for partially observable environments. The proposed idea can be combined with advanced sequence models such as State Space Model. The empirical result on POPGym, a benchmark for memory capability, shows that the proposed method outperforms the widely-used truncated backpropagation through time (BPTT) approach.
Some reviewers initially raised concerns regarding clarity, as they were initially confused or even misunderstood the paper. However, most of the questions were clarified during the rebuttal period, and all of the reviewers ended up leaning towards acceptance. The reviewers acknowledged that the proposed method is novel and promising as a new approach to deal with partial observability in RL. Thus, I recommend to accept the paper. I suggest the authors to revise the paper to make it easier to read for the camera-ready version.